# Mouse embryo geometry drives formation of robust signaling gradients through receptor localization

Zhechun Zhang[1,2,6]*, Steven Zwick [2,6], Ethan Loew [1], Joshua S. Grimley[3,5] & Sharad Ramanathan [1,2,4]*

Morphogen signals are essential for cell fate specification during embryogenesis. Some receptors that sense these morphogens are known to localize to only the apical or basolateral membrane of polarized cell lines in vitro. How such localization affects morphogen sensing and patterning in the developing embryo remains unknown. Here, we show that the formation of a robust BMP signaling gradient in the early mouse embryo depends on the restricted, basolateral localization of BMP receptors. The mis-localization of receptors to the apical membrane results in ectopic BMP signaling in the mouse epiblast in vivo. With evidence from mathematical modeling, human embryonic stem cells in vitro, and mouse embryos in vivo, we find that the geometric compartmentalization of BMP receptors and ligands creates a signaling gradient that is buffered against fluctuations. Our results demonstrate the importance of receptor localization and embryo geometry in shaping morphogen signaling during embryogenesis.

[1] Department of Molecular and Cellular Biology, Harvard University, Cambridge, MA 02138, USA. [2] School of Engineering and Applied Sciences, Harvard University, Cambridge, MA 02138, USA. [3] Allen Institute for Brain Science, Seattle, WA 98109, USA. [4] Department of Stem Cell and Regenerative Biology, Harvard University, Cambridge, MA 02138, USA. [5] Present address: Universal Cells, Seattle, WA 98121, USA. [6] These authors contributed equally: Zhechun Zhang, Steven Zwick. *email: zhechunzhang@fas.harvard.edu; sharad@post.harvard.edu

Morphogens are long-range signaling molecules that move in extracellular space to induce concentration-dependent cellular responses in their target tissues[1,2]. Genetic perturbation of morphogens and their cognate receptors often leads to missing cell types and embryonic structures[3–7]. Many mechanisms have been proposed to explain how morphogens induce signaling gradients in target tissues and therefore direct the spatial organization of cell fates[1,2,8–19]. Surprisingly, several morphogen receptors have been found to localize to either the apical or basolateral membrane of epithelial tissues[15,20–24]. Such localization can dramatically affect how the target tissue senses morphogens[15,20,23]. How receptor localization modulates morphogen signaling in developing embryos is not known.

The early mouse embryo (E6.0–E6.5) adopts an egg-cylinder geometry (Fig. 1a)[5,6,25]. It contains a lumen (the pre-amniotic cavity) encased by two epithelial tissues: the epiblast and extraembryonic ectoderm (ExE). The ExE secretes the morphogen BMP4, which is sensed by receptors in the epiblast[5–7]. The resulting BMP signaling is required for the differentiation of the epiblast into mesoderm[3,4]. Both the epiblast and ExE have stereotyped epithelial tissue geometries[26], with their apical membranes surrounding the lumen and their basolateral membranes facing a narrow interstitial space (between these tissues and the underlying visceral endoderm [VE]). This lumen and interstitial space are separated by impermeable tight junctions present throughout the epithelia except at the border between the ExE and epiblast (Fig. 1a). Indeed, when small-molecule dye fluorescein was injected into the pre-amniotic cavity of an E6.5 mouse embryo, it did not penetrate the epiblast or ExE but diffused through a channel at the edge of the epiblast (Supplementary Fig. 1). Thus, the extracellular space in the embryo through which BMP4 ligands diffuse is compartmentalized into a lumen and an interstitial space.

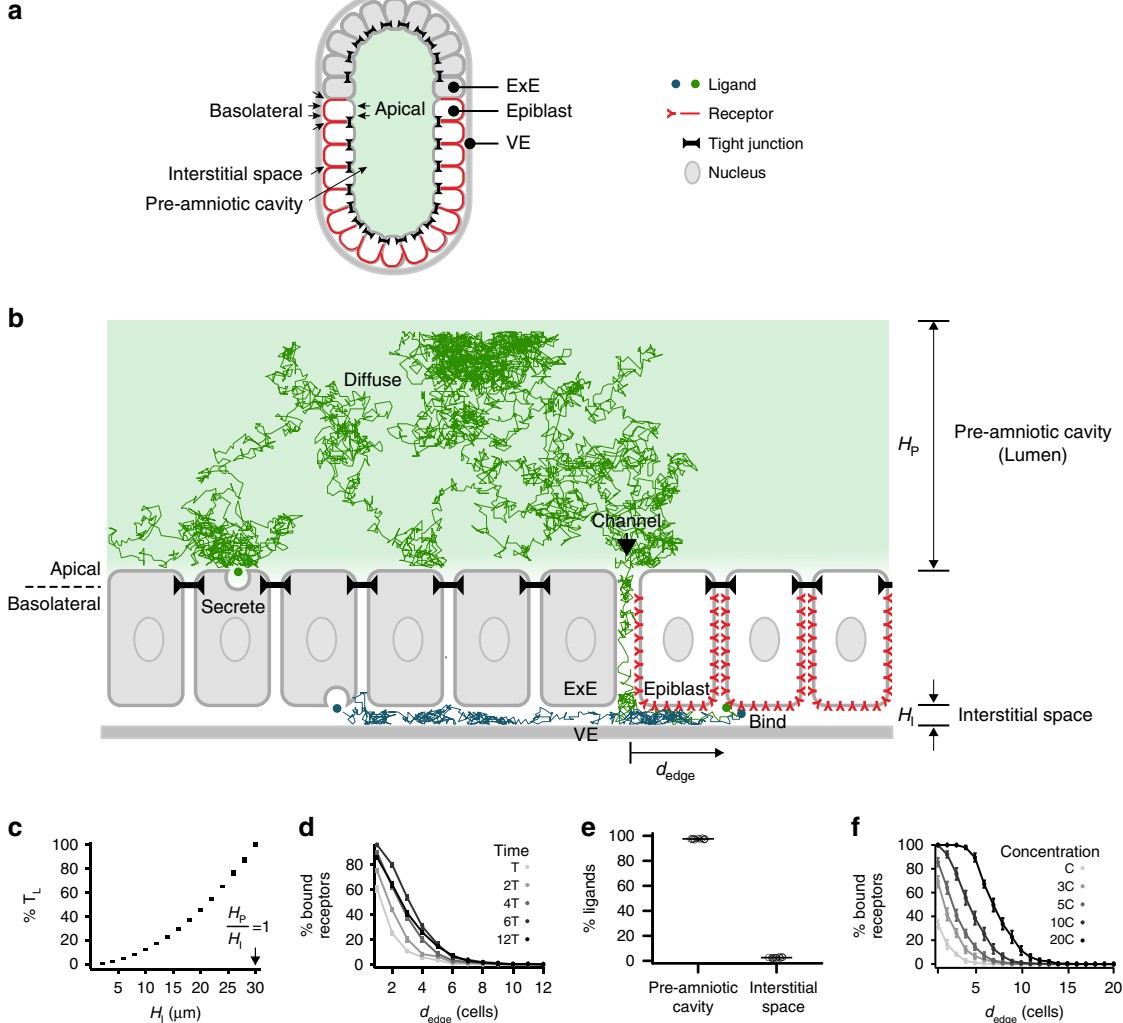

**Fig. 1** Receptor localization facilitates the formation of a robust signaling gradient in early mouse embryo. **a** Illustration of pre-gastrulation mouse embryo, with the epiblast (white) and extraembryonic ectoderm (ExE, light gray) together enclosing the pre-amniotic cavity. Apical membranes of epiblast cells face the pre-amniotic cavity whereas basolateral membranes face the interstitial space. **b** Illustration of a simulation with basolateral receptors. ExE cells secrete BMP4 ligands from their apical (green) or basolateral (blue) membranes, while epiblast cells have BMP receptors (red) on their basolateral membranes. Ligands cannot diffuse past tight junctions between cells (black). Simulated ligand trajectories show that ligands diffuse from epiblast edge (black arrow) through interstitial space to approach and bind basolateral receptors. $H_P$ and $H_I$ denote heights of pre-amniotic cavity and interstitial space, respectively. **c** The time between BMP4 ligands entering interstitial space and being captured by receptors, $T_L$, increases with the height of the interstitial space. **d** Percentage of ligand-bound receptors as a function of their distance from epiblast edge, $d_{edge}$, over time in simulations with apically secreted ligands ($T = 7.5$ min). **e** Percentage of ligands in pre-amniotic cavity vs. interstitial space at steady state (90 min) in simulations with apically secreted ligands. **f** Percentage of ligand-bound receptors as a function of $d_{edge}$ at steady state (90 min) shows signaling gradients at different BMP4 concentrations in simulations with apically secreted ligands ($C = 0.16$ ng/mL or ligand/receptor ratio of 0.1). Error bars denote SEM

Here, by combining mathematical modeling, quantitative imaging, embryological perturbation, and microfluidics, we demonstrate that restricted receptor localization in conjunction with the compartmentalized embryo geometry constrains the diffusion of and therefore response to BMP4 ligands. We show that the BMP signaling gradient arises from the edge of the epiblast even under conditions of uniform BMP4 stimulation. Further, the interplay between restricted receptor localization and the compartmentalized embryo geometry buffers BMP4 ligands in the pre-amniotic cavity through an entropic effect. This entropic buffering renders the formation of BMP signaling gradient robust to fluctuations in BMP4 level. Consistently, mis-localizing BMP receptors in the mouse embryo leads to ectopic BMP signaling. Thus, receptor localization and embryo geometry together play an essential role in regulating morphogen signaling during early development.

## Results

**Receptor localization facilitates the formation of signaling gradient.** To understand how receptor localization impacts BMP signaling between the ExE and epiblast, we simulated the movement of individual BMP4 ligands in the early mouse embryo (E6.0–E6.5) from secretion to receptor binding, using Brownian dynamics[27]. Given the evidence of polarized ligand secretion by epithelial cells in vitro[20,21] (Supplementary Fig. 2), we modeled different instances in which BMP4 ligands were secreted apically (into the pre-amniotic cavity) or basolaterally (into the interstitial space) by the ExE (Fig. 1b). After secretion, ligands diffused through extracellular space in the embryo. Due to tight junctions in the simulation, ligands could move between the pre-amniotic cavity and the interstitial space only by diffusing through the channel between the ExE and epiblast. Some morphogen receptors are known to localize to only the apical or basolateral membranes of epithelial cells[15,21,22,24,28]; such localization could determine the compartment from which ligands are sensed by receptors in the epiblast. Therefore, we also performed simulations with BMP receptors localized exclusively on the apical membrane (facing the pre-amniotic cavity) or basolateral membrane (facing the interstitial space) of epiblast cells. Finally, our model assumed that once BMP4 ligands bound their receptors, signaling activity was induced and the ligands were cleared.

Our simulations show that if the BMP receptors are basolaterally localized in the epiblast, the compartmentalized geometry of the embryo naturally results in the formation of a robust BMP signaling gradient. This occurs despite the absence of other regulatory mechanisms such as signaling inhibitors[2,10–12,15,29]. The basolateral localization of BMP receptors requires that ligands diffuse through the interstitial space between the epiblast and VE to access them (Fig. 1b). The height of this interstitial space, $H_I$, regulates the time, $T_L$, and hence the distance a ligand can diffuse before being captured by a receptor (Fig. 1c). As a consequence, BMP4 ligands are more likely to bind receptors that are closer to the epithelial edge, giving rise to a BMP signaling gradient from the edge of the epiblast inward (Fig. 1d). The signaling gradient forms regardless of whether BMP4 ligands are secreted from the apical or basolateral membrane of the ExE and arises even if ligands are imposed to be uniformly distributed in the pre-amniotic cavity (Supplementary Fig. 3a).

The basolateral localization of BMP receptors, in conjunction with the asymmetric compartmentalization of the embryo, also makes formation of this BMP signaling gradient robust to fluctuations in the BMP4 source strength. Due to the large volume difference between the pre-amniotic cavity and interstitial space and the channel (between the ExE and epiblast) that

connects them, the majority of BMP4 ligands accumulate in the cavity on the apical side of the epiblast (Fig. 1e, Supplementary Fig. 3b). This is an entropic effect: the entropy of BMP4 ligands is maximized when the ligands are uniformly distributed between the pre-amniotic cavity and the interstitial space. In other words, the accumulation of BMP4 ligands in the cavity is driven by the same physical forces that allows ink to diffuse through water and ultimately reach a uniform distribution independent of where ink is dropped initially. Consistently, BMP4 ligands accumulate in the pre-amniotic cavity, regardless of whether the ligands are secreted apically or basolaterally from the ExE in the simulation (Fig. 1e, Supplementary Fig. 3b).

This accumulation results in an entropic buffering effect: the pre-amniotic cavity serves as a ligand reservoir that buffers the signaling gradient against fluctuations in the BMP source strength. Indeed, if the total ligand concentration is increased by tenfold in a simulation with basolateral receptors, the signaling gradient shifts inward by only a few cell widths (Fig. 1f, Supplementary Fig. 3c). However, as the size of the pre-amniotic cavity is reduced in the simulation, increases in ligand concentration shift the signaling gradient significantly further into the epiblast (Supplementary Fig. 4), demonstrating the buffering effect. Strikingly, if BMP receptors are apically localized in the epiblast or if tight junctions are absent, this tenfold increase is sufficient to saturate all receptors in the simulation and destroy the signaling gradient (Supplementary Fig. 3c). Thus, the entropic buffering of the BMP signaling gradient relies upon both the basolateral receptor localization and embryonic geometry in our simulation. Variations in other simulation parameters, such as the ligand diffusion coefficient $D$, the probability of binding between ligand and unbound receptors upon contact $P_{binding}$, and the turnover rate of ligand–receptor pairs $T_t$, do not similarly disrupt the formation of this signaling gradient (Supplementary Figs. 7–9). Likewise, the signaling gradient forms regardless of whether the embryo is rotationally symmetric or if the channel between the ExE and the epiblast is present only at the posterior side in the simulation (Supplementary Fig. 5), even though in the latter case the gradient is more prominent at the posterior side. In summary, our simulations demonstrate that the formation of the signaling gradient is robust in that it can form under wide variety of condition; and further, while the scale of the gradient increase with source strength, this increase is limited by basolateral receptor localization and the asymmetric compartmentalization of the embryo.

Assuming that the BMP receptors are basolaterally localized, our model provides three experimentally testable predictions. First, a BMP signaling gradient will form inward from the epiblast edge even if ligands are present at high concentration throughout the lumen (Supplementary Fig. 3a). Second, formation of this signaling gradient will be robust to fluctuations in BMP concentration (Fig. 1f). Third, the mis-localization of BMP receptors to the apical membrane should lead to ectopic BMP signaling in the epiblast (Supplementary Fig. 10), since apically localized receptors will be able to detect BMP4 ligands that are buffered in the lumen (Fig. 1e).

**Basolateral localization of BMP receptors in hESCs and mouse epiblast.** We asked whether BMP receptors are indeed basolaterally localized in mammalian cells. We measured the localization of these receptors through surface immunostaining[22,24] as well as by imaging GFP- and epitope-tagged receptors (see Methods). The BMP co-receptors BMPR1A (Fig. 2a, b, g–j) and BMPR2 (Fig. 2k, l) are basolaterally localized in human embryonic stem cells (hESCs[15]). We moreover found that the majority of TGF-β family receptors (including BMP receptors) in sequenced

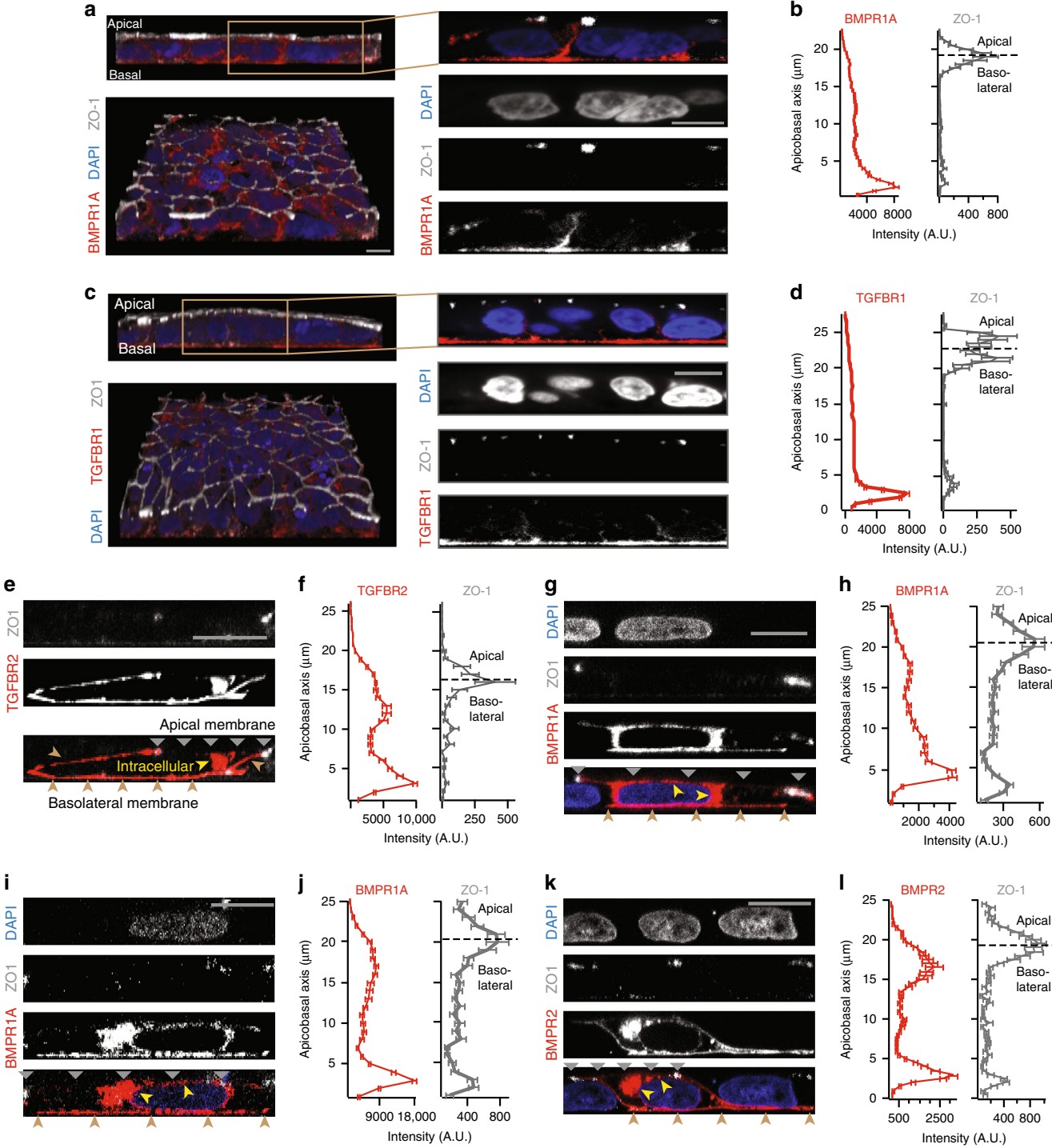

**Fig. 2** BMP and TGF-β receptors localize at basolateral membrane of hESCs in vitro. **a** Left column: 3D confocal images of hESC colony stained for BMPR1A (red), tight junction marker ZO-1 (white), and DNA (blue), in lateral (top) and tilted view (bottom). Right column: Zoomed-in lateral images. Scale bar 10 μm. **b** Plots of BMPR1A (left) and ZO-1 (right) staining intensity along apicobasal axis show BMPR1A localized beneath tight junctions ($n = 38$ cells from two experiments). **c** Left column: 3D confocal image of hESC colony stained for DNA (blue), TGFBR1 (red), and ZO-1 (white) in lateral (top) and tilted view (bottom). Right column: Zoomed-in lateral section. **d** Plots of TGFBR1 (left) and ZO-1 levels (right) against apicobasal axis show that TGFBR1 is localized below tight junctions ($n = 51$ cells from two experiments). **e** Confocal image of a hESC expressing TGFBR2-Clover (red), stained for ZO-1 (white). **f** TGFBR2-Clover (left) and ZO-1 levels (right) against apicobasal axis ($n = 4$ cells from two experiments). **g**, **h** Same as **e**, **f** but for BMPR1A-Clover ($n = 2$ cells from two experiments). **i**, **j** Same as **e**, **f** but for BMPR1A-HA ($n = 3$ cells from two experiments). **k**, **l** Same as **e**, **f** but for BMPR2-Clover ($n = 3$ cells from two experiments). Yellow arrows in **e**, **g**, **i**, **k** denote intracellular receptors in secretory pathway. Gray and brown arrows indicate apical and basolateral membranes, respectively. Scale bar 10 μm. Error bars denote SEM

**Fig. 3** LTA motif in TGF-β superfamily receptors. Protein sequence alignment of TGF-β superfamily receptors shows conservation of LTA motif in nine receptors and a LSA motif with a conservative substitution of serine for threonine in two receptors

vertebrates contain a conserved LTA amino acid motif near their C-terminus (Fig. 3). This motif has been shown to be necessary and sufficient for the basolateral localization of TGFBR2 in MDCK cells, and the mutation of this motif to an LTG sequence leads to the receptor's mis-localization to the apical membrane[22]. Consistently, we found that TGFBR2 and its co-receptor TGFBR1 are localized at the basolateral membrane of epithelial human hESCs (Fig. 2c–f). Furthermore, the ACTIVIN/NODAL receptors ACVR1B and ACVR2B have also been found to be basolaterally localized in studies using human gastruloids[15], consistent with

the fact that these receptors have LTA motifs (Fig. 3). Thus, an evolutionarily conserved LTA motif is present in all of these receptors that are exclusively localized along the basolateral membrane in hESCs.

We next explored whether BMP receptors are similarly localized in the basolateral membrane of mouse epiblast cells in vivo. To visualize receptors specifically on the cell membrane, we developed a protocol for surface immunostaining the mouse epiblast around the start of gastrulation (see Methods). After collection of E6.5 mouse embryos, we surgically removed the ExE

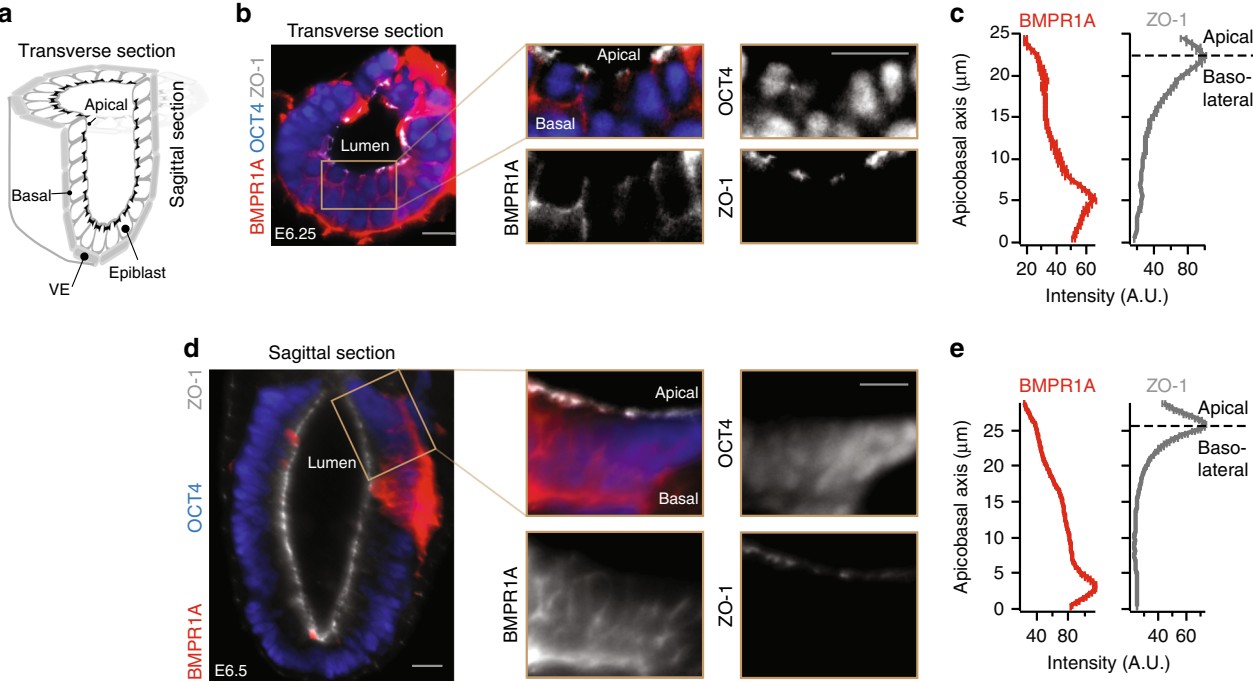

**Fig. 4** BMP receptors localize at basolateral membrane of mouse epiblast in vivo. **a** Illustration of pre-gastrulation mouse embryo, showing transverse and sagittal sections. **b** Transverse section of an E6.25 mouse embryo stained for epiblast marker OCT4, BMPR1A, and ZO-1. **c** Plots of BMPR1A (left) and ZO-1 (right) staining intensity along apicobasal axis for transverse section from **b** show BMPR1A localized beneath tight junctions. **d, e** Same as in **b, c** but for a sagittal section of an E6.5 mouse embryo. These images are representative of two sets of images in two embryos/experiments. In mouse data, scale bar 20 μm. Error bars denote SEM

from each embryo and exposed the epiblast to BMPR1A antibodies. We subsequently fixed and permeabilized the embryos and immunostained them for tight junction protein ZO-1 and epiblast marker OCT4. Light-sheet microscopy of the immunostained embryos shows that BMPR1A receptors in epiblast cells are localized on the basolateral membrane facing the underlying VE (Fig. 4, Supplementary Fig. 11).

**A robust BMP signaling gradient forms from the epiblast edge.** We asked whether the predicted formation of a robust BMP signaling gradient would occur in the epiblast. We first measured the distribution of phosphorylated SMAD1/5 (pSMAD1/5, the downstream effectors of the BMP signaling pathway) in epithelial hESC colonies exposed to BMP4 ligands. These epithelial colonies have impermeable tight junctions and a narrow, permeable basement membrane matrix underneath mimicking an interstitial space. The tissue geometry therefore is comparable to the geometry of the epiblast in mammalian embryos[25,30]. Akin to the simulation, we observed pSMAD1/5 gradients organized from the edges of epithelial hESC colonies exposed to spatially uniform concentrations of BMP4 (Fig. 5a, b, Supplementary Fig. 12a, b). The formation of these BMP signaling gradients were robust to changes in ligand concentration: colonies exposed to BMP4 concentrations across a 1000-fold range displayed stable pSMAD1/5 gradients inward from colony edges, with the depth of the gradient varying only between 2 and 10 cell widths (Fig. 5c, d and Supplementary Fig. 12e). This limited change in depth is consistent with our simulation results (Supplementary Fig. 13). The formation of these robust gradients was dependent on the segregation of apical and basolateral extracellular compartments by tight junctions. When tight junctions were disturbed by a brief treatment of passaging reagent ReLeSR or calcium chelator EGTA[31] before BMP4 induction, signal response occurred throughout hESC colonies (Supplementary Fig. 12c).

Further, if hESCs were exposed to uniform BMP4 shortly after single-cell passaging, cells that had not yet formed tight junctions with adjacent cells showed significantly higher pSMAD1/5 activity than those surrounded by tight junctions (Supplementary Fig. 12d).

We observed similar BMP signaling gradients in early mouse embryos as well. In harvested mouse embryos stained for pSMAD1/5, we observed a gradient of pSMAD1/5 activity inward from the proximal edges of the epiblast at the pre-streak (~E6.25) through the early streak (~E6.75) stages of development (Fig. 5e, f, Supplementary Fig. 6). To test whether this signaling gradient is maintained even in uniformly high concentrations of BMP4, we surgically removed the ExE from E6.5 mouse embryos, exposing the remaining epiblast-VE cup. We then soaked the cup in media containing 10 ng/mL BMP4 for 30 min before fixing and immunostaining for pSMAD1/5 (Fig. 5g). In these BMP-soaked embryos, the pSMAD1/5 gradient reached only a few cell widths further from the proximal epiblast edge as compared to wild-type embryos (Fig. 5g, h). This restriction of BMP signaling was maintained despite the fact that the BMP4 concentration was sufficiently high to induce pSMAD1/5 activity uniformly throughout the epiblast if its basolateral surface was exposed to ligands (Supplementary Fig. 12f). In summary, our results in vitro and in vivo show that gradients of BMP signaling activity robustly form inward from the edges of epithelial tissues with basolateral receptor localization.

**Mis-localization of receptors leads to ectopic BMP signaling.** Having verified the first two predictions of the model, we next tested whether the mis-localization of BMP receptors to the apical membrane results in ectopic BMP signaling. To do so, we designed a plasmid expressing epitope-tagged mutant copies of both *BMPR1A* and *BMPR2*, in which their LTA motifs were mutated into an LTG sequence (see Methods). Unlike the

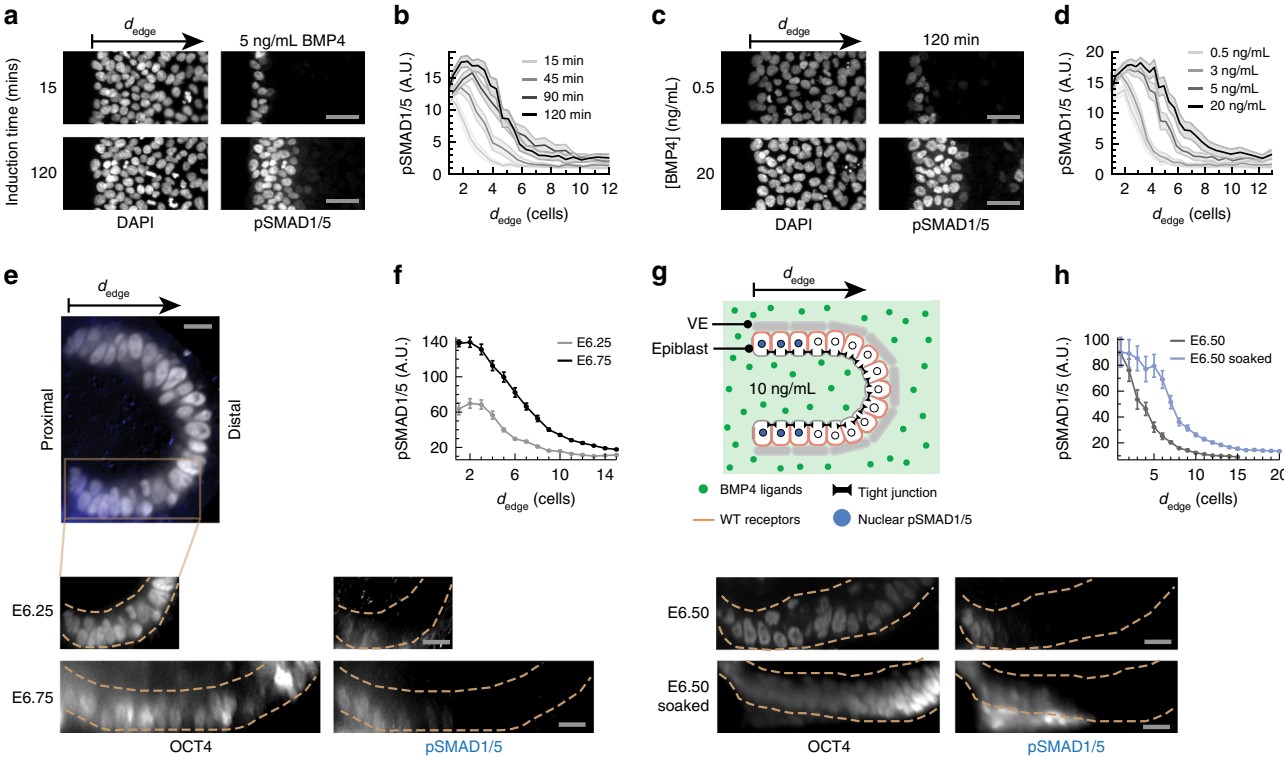

**Fig. 5** A robust BMP signaling gradient forms from the epiblast edge. **a** hESC colony stained for DNA and pSMAD1/5 after 15 or 120 min BMP4 induction. These images are representative of five sets of images from two experiments. **b** pSMAD1/5 level of single hESCs as a function of their distance from the nearest colony edge, or $d_{edge}$, after 15–120 min of BMP4 induction. **c** hESC colony stained for DNA and pSMAD1/5 after 120 min exposure to 0.5–20 ng/mL BMP4. These images are representative of five sets of images from two experiments. **d** pSMAD1/5 level of single hESCs as a function of $d_{edge}$ after 120 min exposure to 0.5–20 ng/mL BMP4. Shaded error bars denote 95% confidence intervals (paired $t$-test). In hESC data, scale bar 50 µm. **e** E6.25–E6.75 mouse embryos stained for OCT4 (white) and pSMAD1/5 (blue). Dotted yellow lines indicate epiblast boundary. **f** pSMAD1/5 level of mouse epiblast cells as a function of their distance from the posterior proximal edge of the epiblast, or $d_{edge}$, for E6.25 (two embryos per experiment) and E6.75 embryos (two embryos per experiment). **g** Top: Illustration of BMP4 exposure experiment. ExE is surgically removed, and remaining epiblast-VE cup is soaked in media containing 10 ng/mL BMP4 for 30 min. Bottom: Intact E6.5 mouse embryo and BMP4-soaked E6.5 mouse embryo, both stained for OCT4 and pSMAD1/5. Dotted yellow lines indicate epiblast boundary. **h** pSMAD1/5 intensity of epiblast cells as a function of $d_{edge}$ for intact (three embryos per experiment) and BMP4-exposed E6.5 embryos (two embryos per experiment). In mouse data, error bars denote SEM and scale bar 20 µm

wild-type receptors, these mutant receptors localized at both the apical and basolateral membranes of hESCs transfected with these plasmids (Fig. 6a, b, Supplementary Fig. 14a). The transfected hESCs, in the absence of exogenous BMP4 ligands, did not show any significant BMP signaling activity (Fig. 6c, Supplementary Fig. 14b). After BMP4 exposure, however, cells expressing the mis-localizing receptors had significantly higher levels of nuclear pSMAD1/5 than their neighboring non-transfected cells (Fig. 6b, c, Supplementary Fig. 14a). The pSMAD1/5 levels of these transfected cells were comparable to that of non-transfected cells at colony edges (Fig. 6c). In contrast, overexpression of wild-type receptors did not lead to a comparable increase in pSMAD1/5 levels of transfected cells in vitro, as predicted by our simulation (Supplementary Figs. 10 and 15). Thus, while basolaterally localized wild-type BMP receptors in the interior of hESC colonies were insulated from apical ligands by tight junctions, cells with mis-localized BMP receptors could sense and respond to these ligands.

To test the effect of receptor mis-localization in vivo, we developed a method to deliver our mutant BMP receptor plasmid to anterior and distal regions of the epiblast that do not normally show BMP signaling activity, while leaving the rest of the mouse embryo unperturbed (see Methods, Supplementary Fig. 14c). Consistent with our results in hESCs, mutant BMP receptors were localized at both the apical and basolateral membranes of transfected epiblast cells in vivo (Fig. 6d). This mis-localization

led to ectopic BMP signaling in cells in the anterior and distal regions of the epiblast, where neighboring non-transfected cells showed no signal response (Fig. 6d, e, Supplementary Fig. 14d). pSMAD1/5 levels in electroporated cells resembled that of cells at the epiblast edge (Fig. 6e). These data support our simulation results, in which BMP4 ligands can be present throughout the pre-amniotic cavity while basolateral BMP receptors in the epiblast are insulated from these signals.

**Distance from tissue edge governs patterning of epithelial tissues**. In summary, our results in silico, in vitro, and in vivo demonstrate how basolateral receptor localization and embryo geometry together, through an entropic buffering mechanism, result in the formation of robust BMP signaling gradients at tissue edges. Consistently, our mathematical model argues that an epithelial cell's distance from the tissue edge ($d_{edge}$) predicts the cell's signaling response better than its distance from the source of the signal ($d_{source}$, Fig. 7a, b). Here, the predictive power is quantified by the proficiency (the mutual information shared between the coordinate of a cell and its pSMAD1/5 levels, given as a percentage out of the total information entropy of pSMAD1/5 levels[32]). While studies in multiple model organisms have shown that $d_{source}$ is a critical determinant of patterning[1,8,9,13,33], our results argue that $d_{edge}$ could also be an important developmental coordinate for the patterning of epithelial tissues.

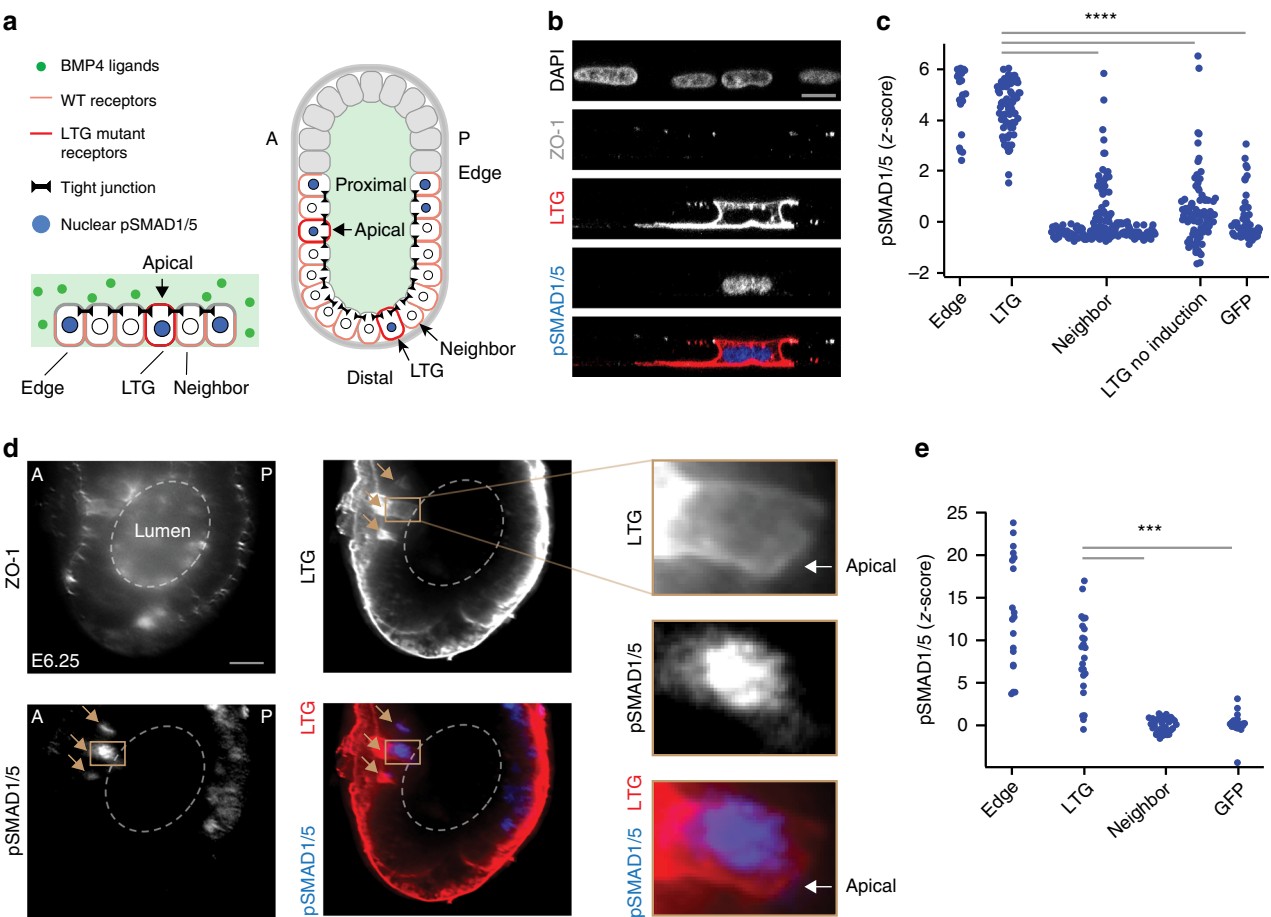

**Fig. 6** Mis-localization of receptors leads to ectopic BMP signaling. **a** Left: Illustration of a hESC colony containing a single cell with mis-localized LTG mutant BMP receptors, exposed to BMP4 ligands. Right: Illustration of mouse embryo with two cells expressing mis-localized LTG mutant BMP receptors, leading to ectopic pSMAD1/5 activity. A and P denote anterior and posterior, respectively. **b** Confocal image of hESC colony transfected with mutant receptor plasmid (BMPR1A$^{A514G}$-Clover-IRES-BMPR2$^{A494G}$), immunostained for pSMAD1/5 after a 30 min BMP4 induction. From top: DNA, ZO-1, Clover (LTG mutant receptors), pSMAD1/5, and color-combined image. Scale bar 10 μm. **c** pSMAD1/5 intensities of hESCs after 30 min BMP4 induction: cells at edge of colony (Edge, $n = 23$ from two experiments); non-edge cells expressing mutant BMPR1A$^{A514G}$ and BMPR2$^{A494G}$ receptors (LTG, $n = 73$ from two experiments); non-transfected neighbors of transfected cells (Neighbor, $n = 166$ from two experiments); cells expressing mutant receptors but without BMP4 induction (LTG no induction, $n = 96$ from two experiments); and cells transfected with GFP plasmid ($n = 51$ from two experiments). Z-score denotes number of standard deviations beyond background mean (of neighboring non-transfected cells), $^{****}p < 0.0001$ (paired $t$-test). **d** E6.25 mouse embryo transfected with mutant receptor plasmid (BMPR1A$^{A514G}$-Clover-IRES-BMPR2$^{A494G}$), immunostained for ZO-1, Clover (LTG mutant receptors), and pSMAD1/5. Images show localization of mutant receptors at both apical and basolateral membrane and pSMAD1/5 activity in a transfected cell. Brown arrows indicate transfected cells. Scale bar 20 μm. **e** pSMAD1/5 intensity epiblast cells: cells at edge of epiblast (Edge, $n = 42$ from 13 embryos); non-edge cells transfected with mutant receptor plasmid (LTG, $n = 27$ from 13 embryos); their neighboring non-transfected cells (Neighbor, $n = 52$ from 13 embryos); and cells transfected with a GFP plasmid (GFP, $n = 20$ from 3 embryos). Edge and LTG cells with z-score greater than 25 are not shown ($n = 26$), $^{***}p < 0.001$ (paired $t$-test)

To test how epithelial cell fate decisions are organized along $d_{source}$ and $d_{edge}$, we developed microfluidic devices capable of producing precise morphogen gradients (Fig. 7c, Supplementary Fig. 16a–c). The environment within the device mimics that of a morphogen gradient produced by a signal source at the left end of the device. We exposed hESC colonies to a BMP4 gradient from 10 to 0 ng/mL for 30 min. Consistent with our previous results (Fig. 5, Supplementary Figs. 12a, b and 17), signaling activity depended strongly on $d_{edge}$ (Fig. 7c, d, Supplementary Fig. 16d). In fact, a cell's $d_{edge}$ had a significantly higher proficiency than $d_{source}$ in predicting its signaling response to the BMP gradient (Fig. 7d, Supplementary Fig. 19a–f).

To determine how $d_{source}$ and $d_{edge}$ correlate with cell fate decision dynamics, we built a dual-color OCT4-RFP SOX2-YFP hESC line, in which OCT4 and SOX2 are tagged with fluorescent proteins at their endogenous loci (Supplementary Fig. 18). OCT4

and SOX2 are co-expressed in the pluripotent state (OCT4+ and SOX2+) but are differentially regulated during mesodermal differentiation (OCT4+ and SOX2−); this differential regulation is essential for the cell's germ layer fate choice[34]. We then cultured epithelial colonies of this hESC line in the microfluidic device, exposing them to gradients of BMP4 and NODAL-analog ACTIVIN A (from 10 to 0 ng/mL, of each). We measured the OCT4 and SOX2 levels of individual cells in these gradients as well as their $d_{source}$ and $d_{edge}$ for 18 h using time-lapse microscopy (Fig. 7e, f, Supplementary Fig. 16e). At the end of the time-lapse, we immunostained the cells in situ for mesodermal progenitor marker BRACHYURY/T to determine their fate choice (Fig. 7e, f, Supplementary Fig. 16f).

We found that cells with comparable $d_{source}$ but different $d_{edge}$ often adopted distinct cell fates (Fig. 7e, f, Supplementary Fig. 16g). In many cases, cells near colony edges had higher

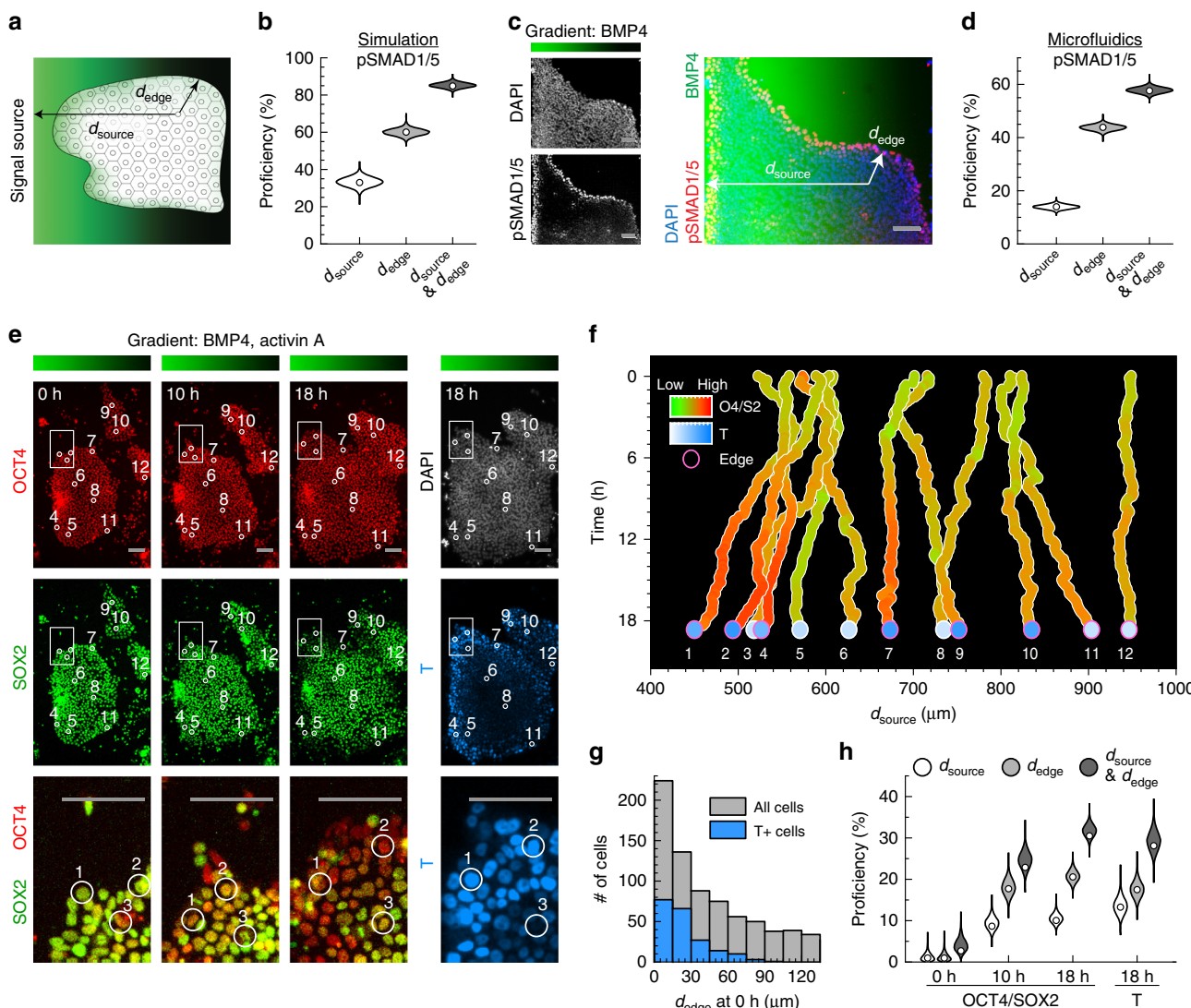

**Fig. 7** Distance from tissue edge and distance from signal source govern patterning of epithelial tissue. **a** Illustration of epithelial tissue within a morphogen gradient emanating from a source to the left. The coordinates $d_{edge}$ and $d_{source}$ denote a cell's distance from the nearest tissue edge and from the signal source, respectively. **b** Proficiency of $d_{source}$, $d_{edge}$, or both coordinates to predict pSMAD1/5 level of epiblast cells in simulation (Fig. 1). **c** Epithelial hESC colony exposed to BMP4 gradient in microfluidic device for 30 min and stained for DNA and pSMAD1/5. BMP4 gradient ranges from 10 ng/mL (green) to 0 ng/mL (black). Coordinates $d_{source}$ and $d_{edge}$ are depicted for a single cell as in **a**. These images are representative of five sets of images from two experiments. **d** Proficiency of $d_{source}$, $d_{edge}$, or both coordinates to predict pSMAD1/5 levels in hESCs exposed to microfluidic BMP4 gradient ($n = 13,828$ cells). **e** OCT4-RFP (red) SOX2-YFP (green) double reporter hESCs within microfluidic gradient after 0, 10, and 18 h of differentiation. DNA, white; BRACHYURY/T, teal. Inset highlights differentiation at colony edges. Position of 12 sample cells labeled by circles. Bar above shows BMP4 and ACTIVIN A gradient within microfluidic device, ranging from 10 ng/mL (green) to 0 ng/mL (black) of each. These images are representative of two sets of images from two experiments. **f** $d_{source}$ of 12 tracked cells from **e** throughout time-lapse, colored by OCT4/SOX2 ratios (red/green) and BRACHYURY/T level (teal) at end of time-lapse. Pink circles mark cells with $d_{edge}$ less than 52 µm at 10 h of differentiation. **g** Distribution of $d_{edge}$ at start of differentiation, with teal marking cells that were BRACHYURY/T+ after 18 h of differentiation. **h** Proficiencies of $d_{source}$, $d_{edge}$, or both coordinates to predict OCT4/SOX2 ratios and BRACHYURY/T levels ($n = 1275$ cells). Violin plots denote Gaussian kernel density estimates. Scale bar 100 µm

BRACHURY/T and lower SOX2 levels than cells in colony interiors that had a smaller $d_{source}$ throughout the time-lapse. Furthermore, 95% of cells that expressed BRACHYURY/T at the end of time-lapse were initially located near colony edges ($d_{edge} < 66.5$ µm or approximately 5.1 cell widths, Fig. 7g), where signaling is most active at the start of differentiation (Fig. 7c). After 48 h of exposure to BMP4 and ACTIVIN A gradients, hESCs with high BRACHYURY/T and low SOX2 levels continued to be located predominantly at the colony edges, while cells in colony interiors remained undifferentiated (Supplementary Fig. 16h, i). Like pSMAD1/5, the dependence of BRACHYURY/T levels on $d_{edge}$ also required epithelial integrity. If hESC colonies were treated

with ReLeSR during the first 8 h of differentiation, cells in colony interiors also had high BRACHYURY/T levels (Supplementary Fig. 16j).

These data argue that the organization of BMP signaling inward from epithelia tissue edges has significant implications for cell fate decisions. Indeed, we found that $d_{source}$ and $d_{edge}$ each carried independent information about cells' fate choices in the microfluidic device (Fig. 7h, Supplementary Fig. 19g–i). Furthermore, the $d_{edge}$ of hESCs had a significantly higher proficiency of predicting their OCT4/SOX2 and BRACHYURY/T levels than their $d_{source}$, demonstrating the importance of a cell's distance from epithelial edges as a developmental coordinate.

## Discussion

Our results identify that the interplay between receptor localization and embryo geometry leads to the formation of a robust BMP signaling gradient. Specifically, the compartmentalized geometry of the early mammalian embryo requires BMP4 ligands to diffuse through a narrow interstitial space to approach basolateral receptors. This constraint limits the time and distance ligands can travel before being captured by receptors, which are spatially restricted. As a result, a signaling gradient naturally arises, even when ligands are present uniformly in the lumen on the apical side of the epiblast. Furthermore, through a geometry-related entropic effect, BMP4 ligands accumulate in the apical lumen. Consequently, this lumen serves as a reservoir that buffers the signaling gradient against fluctuations in BMP4 concentration. Due to this entropic buffering mechanism, the channel between the ExE and epiblast that connects the apical lumen and basal interstitial space acts like a stable BMP4 source for the epiblast. Therefore, a robust BMP signaling gradient forms spontaneously due to the compartmentalized embryo geometry and basolateral receptor localization. While receptor localization has been shown to regulate morphogen signaling in cells in vitro[15,21,23,28] and adult tissues[20], this study demonstrates the effects of receptor localization on morphogen signaling in the developing embryo.

Our current model neglects the possible effects of other regulators of BMP signaling, such as BMP activators and inhibitors. In particular, TGF-β family inhibitors LEFTY1 and CER1 are expressed in the anterior VE of the mouse embryo at E5.75 where they are required for proper patterning during gastrulation[35]. We anticipate that inclusion of such regulators to the model would restrict BMP signaling more to the posterior edge of the epiblast and could contribute further robustness to the BMP signaling gradient against fluctuations in ligand concentration[5,6,10,12,14,15,36]. Nevertheless, our results show that embryonic geometry and receptor localization are sufficient to produce robust gradients of BMP signaling and to explain how mis-localization of BMP receptors leads to ectopic signaling in anterior and distal epiblast cells (Fig. 6). It would be particularly interesting to incorporate BMP regulators in future versions of the model given that they too can be constrained by the compartmentalization of the embryo[15,37].

This study leads to the question of how access to BMP4 ligands changes as the geometry of the embryo rapidly transforms during gastrulation. As a consequence of BMP signaling, epiblast cells undergo epithelial–mesenchymal transitions (EMT) and ingress within the primitive streak[3–6]. Our results suggest that the breaking of tight junctions during EMT may create additional channels between the pre-amniotic cavity and interstitial space, allowing BMP4 ligands to reach previously inaccessible receptors near sites of ingression. Therefore, our results indicate a possible feedback loop between embryo geometry and signaling, in which the epithelial integrity of the epiblast restricts BMP signaling while BMP signaling promotes breakdown of the epithelium. How this geometric feedback, in conjunction with the mechanical forces present during EMT[38–40], regulates the distal extension of the primitive streak and the patterning of the streak fates along the anterior–posterior axis is an important subject of future investigation.

We expect that our mechanism of signaling gradient formation will be broadly applicable in many developmental contexts. Epithelial tissues naturally compartmentalize the extracellular space of developing embryos[25,26,41], as found elsewhere in the migrating zebrafish lateral line primordium[41] and the imaginal wing disc of flies[42]. In particular, while the shape of the epiblast differs among mammalian species, its role in compartmentalizing the pre-gastrulation embryo into an apical lumen and basolateral interstitial space is conserved[6,25,30,43]. The observation that TGF-β family receptors in species ranging from flies to mammals contain the LTA amino acid motif suggests that their basolateral localization may also be evolutionarily conserved. It has been shown in a concurrent study that mutation of the LTA motif results in apical mis-localization of the TGF-β receptor in the polarized intestinal cells of *C. elegans*[44], supporting this hypothesis. Moreover, the LTA motif of human TGF-β receptors overlaps with several germline missense mutations associated with Marfan-like syndromes[44], indicating the potential importance of receptor mis-localization for developmental diseases.

Our results demonstrate how the combination of compartmentalization and receptor localization restricts the sensing of morphogens in developing tissues, which can dramatically modulate signaling and downstream tissue patterning. Therefore, future studies should take these factors into account when considering how morphogen signals pattern the embryo during development.

## Methods

**Simulation of BMP4 dynamics.** In 2D simulation, BMP4 ligands are secreted by six ExE cells and received by 20 epiblast cells arranged linearly along the proximal–distal axis. The 3D simulation contains 20 such linearly arranged arrays of cells in parallel along the anterior–posterior axis. As a result, the 3D simulation contains 120 ExE cells and 400 epiblast cells. Periodic boundary condition was applied along the anterior–posterior axis. Each cell is 8 μm wide and 18 μm tall. The pre-amniotic cavity above the cells is 260 μm wide and 30 μm tall. The interstitial space is 260 μm wide and 2 μm tall. The lateral separation between cells is 2 μm. The simulation setup is therefore comparable to the geometry of the pre-gastrulation mouse embryo (300 μm in length and 100 μm in width). The height of the interstitial space and lateral separation between cells were estimated from fluorescein injection experiments and images of embryos stained for BMPR1A. For 3D simulations of BMP signaling in hESC colonies, we removed ExE cells from the simulation and used the same parameters.

Each epiblast cell has 100 receptors. In 2D simulation, by default 1000 ligands are initially secreted uniformly from the ExE at either the apical membrane or the basal membrane. Although the true number of ligands and receptors may likely be different in the mouse embryo, our simulation results hold for a wide range of scenarios, from the regime where ligands (4000) heavily outnumber receptors (2000) to the regime where receptors (2000) heavily outnumber ligands (200). 3D simulation, in comparison, contains 40,000 receptors and 4000–80,000 ligands.

After secretion, BMP4 ligand diffusion is simulated as a random walk following Brownian dynamics. Ligand positions are updated after each time step $h$ according to the equation $\vec{r}_i(t + h) = \vec{r}_i(t) + \vec{\Gamma}_i(t)$, where $\vec{r}_i(t)$ is the position of ligand $i$ at time $t$. $\vec{\Gamma}_i(t)$ is a random Brownian force acting on ligand $i$ that satisfies constraints $\vec{\Gamma}_i(t) = 0$ and $\vec{\Gamma}_i(t)\vec{\Gamma}_j(t') = cDh\delta_{ij}\delta_{tt'}$, where $D$ is the diffusion coefficient and $c = 4$ for 2D simulations or $c = 6$ for 3D simulations[27]. We estimate $D = 20$ μm²/s by default based on diffusion measurements of BMP homolog Dpp in the larval wing disc of *Drosophila melanogaster*[18]. We use the "local" diffusion coefficient measured by fluorescence correlation spectroscopy rather than the "global" diffusion coefficient measured by FRAP since our simulation explicitly models ligand–receptor binding, which has been shown to slow ligand diffusion at larger length scales[2,18]. Each simulation integration step occurs after $h = 3$ ms.

The diffusing ligands are not allowed to diffuse through tight junctions between cells, cell membranes, or the outer boundaries of the pre-amniotic cavity and interstitial space. Incoming ligands are instead reflected at these surfaces. Given that tight junctions are absent between the ExE and the epiblast, ligands in the pre-amniotic cavity are allowed to reach the interstitial space, and vice versa, through the gap at the edge of epiblast.

If a ligand contacts an epiblast cell membrane that has any unbound receptors, the ligand binds the receptor with probability $P = P_{binding}R_{unbound}$, where $P_{binding}$ is the probability a ligand binds a nearby unbound receptor and $R_{unbound}$ is the fraction of receptors on the membrane that are not bound by ligand. By default, $P_{binding} = 0.002$. For both 2D and 3D simulations, each epiblast cell has 80 receptors on lateral membrane and 20 receptors on its basal membrane.

After $T_t = 45$ min, a timescale related to the endocytosis and recycling of ligand-bound receptors[17,45,46], each receptor–ligand pair is replaced by an unbound receptor on the same epiblast cell membrane and an unbound ligand secreted by the same ExE cell. This coupling between releasing of unbound receptor and unbound ligand was to maintain the total ligand concentration as a constant. As a control, we also performed simulations in which (a) no unbound ligand was secreted by ExE cell upon endocytosis of ligand–receptor pair and total ligand concentration is slowly decreasing over time (b) unbound ligands were constantly secreted by ExE cells and total ligand concentration is slowly increasing over time.

For any given set of parameters, simulations were repeated 10 times. The simulation had no other parameters and was coded in C. The code was commented

and available at 10.6084/m9.figshare.9684992. To run the code: (1) compile the C-code in terminal (on Mac OSX or cluster) by: g++-o test.exe comment2-loop-simulp3d7tov.cpp. (2) Run the exe by: ./test.exe.

A particle diffusion simulation was utilized rather than a reaction-diffusion model to study the effects of (i) embryonic geometry (Supplementary Fig. 4), (ii) polarized ligand secretion (Supplementary Fig. 2), (iii) receptor mis-localization (Supplementary Fig. 10) on BMP signaling with an intuitively understandable approach.

Although direct quantitative comparison between the model and experiment is not possible without precise knowledge of biochemical parameters, we expect our model to agree with experiment qualitatively in the following five criteria: (i) pSmad1/5 as a function of time (Figs. 1d and 3b, f); (ii) pSmad/15 as a function of concentration (Figs. 1f and Fig. 3d, h, Supp Fig. 14), (iii) when tight junctions are broken (Supp Figs. 2c and 13c, d), (iv) when receptors are mis-localized (Fig. 4, Supp Figs. 2c and 10), (v) mutation information between pSMAD1/5 and distance from epithelial edge (Fig. 5b, d).

**Cell lines used in the study**. All hESC experiments were performed with WA01 (H1) cells or SOX2-YFP, OCT4-RFP double reporter cells (see below) in an H1 background.

**Cell culture and passage**. hESCs were maintained in the feeder-free cell culture medium mTeSR1 (STEMCELL Technologies) with daily media changes. For passaging, cells were dissociated en bloc with ReLeSR (STEMCELL Technologies) following the manufacturer's protocol, and detached ES cell clumps were broken into smaller pieces (10–20 cells) by tapping the plate or gently pipetting several times with a wide-bore P1000 micropipette (Corning). Cells were passaged at a 1:12 split ratio onto Matrigel-coated (Corning) plates. Immediately following passage, cells were maintained in mTeSR1 supplemented with 10 μM ROCK inhibitor Y-27632 (STEMCELL Technologies) for 24 h before returning to culture in mTeSR1 alone.

**Surface immunostaining of hESCs**. Before surface receptor staining[21], cells were rinsed once in 1× PBS (Lonza). Cells were incubated with primary antibodies diluted in mTeSR1 with 1% BSA and 5% normal donkey serum at 37 °C for 45 min. Afterward, cells were rinsed two times in PBS and subsequently fixed in 4% formaldehyde for 20 min at room temperature. Secondary stains were then performed (see Intracellular immunostaining of hESCs).

**Intracellular immunostaining of hESCs**. Cells were fixed for 20 min at room temperature in 4% formaldehyde and rinsed three times with PBS. Permeabilization and blocking were performed simultaneously by incubating cells in blocking buffer (PBS with 5% normal donkey serum and 0.3% Triton X-100) for 1 h at room temperature. Primary antibody incubation was performed overnight at 4 °C in antibody dilution buffer (PBS plus 1% BSA, and 0.3% Triton X-100). The next day, cells were washed with PBS three times and then incubated with DAPI and secondary antibodies in antibody dilution buffer (as above) for 1 h at room temperature. After secondary stain, cells were washed with PBS three times before imaging.

**Antibodies**. BMPR1A (1:20, sc20736; Santa Cruz); BRACHYURY/T (1:400, AF2085; R&D); Clover (1:600, EMU101; Kerafast); OCT4 (1:800, sc8628; Santa Cruz); pSMAD1/5 (1:800, 13820s; Cell Signaling); TGFBR1 (1:20, sc9048; Santa Cruz); ZO-1 (1:100, 33-9100; Thermo Fisher); ZO-1-FITC (1:100, 33-9111; Thermo Fisher).

**Plasmid construction and transient expression of receptors**. Receptor genes (BMPR1A and BMPR2) were cloned into the plasmid pCAGIP-TGFBR2-Clover (a gift from the Jeff Wrana lab at Lunenfeld Tanenbaum Research Institute) between restriction sites XhoI and AgeI. To visualize receptors using small epitope tags, Clover was replaced by an Myc or HA tag between restriction sites AgeI and NotI. To minimize side effects caused by plasmid expression of tagged protein, we excluded cells with excessive levels of expression, aggregates of fluorescent proteins, and membrane blebbing from downstream analysis.

**Plasmid construction and receptor mis-localization**. To mis-localize receptors, LTA motifs in both BMPR1A and BMPR2 were mutated into LTG sequences[22] in our plasmids by site-directed mutagenesis (NEB). The puromycin in the pCAGIP-BMPR1A-Clover plasmid was replaced by BMPR2-Myc between restriction sites BmgBI and SacI. To minimize side effects caused by plasmid expression of tagged protein, we excluded cells with excessive levels of expression, aggregates of fluorescent proteins, or membrane blebbing from downstream analysis.

**hESC transfection**. Transfection of hESCs was performed using jetPrime (Polyplus-transfection) or the Amaxa Nucleofector II (Lonza). For jetPrime transfection, hESCs were transfected within 2 days of passage following the manufacturer's protocol. For nucleofection, hESC cell colonies were dissociated into single cells

(see Single-cell passaging) and split into aliquots of 800,000 cells. Aliquots were spun for 3 min at $200 \times g$ before resuspension in 82 μL human stem cell Nucleofector Solution 2 (Lonza) and 18 μL Supplement 1 (Lonza) with 1–5 μg of DNA. The cell suspension was added to a nucleofection cuvette, and transfection was carried out using nucleofection program B016. Immediately following transfection, 500 μL of mTeSR1 culture medium (STEMCELL Technologies) supplemented with 10 μM ROCK inhibitor (STEMCELL Technologies) was added to the cuvette, and cells were seeded into a 15 mm well (Corning) coated with Matrigel (Corning).

**Breaking tight junctions**. hESC colonies were washed once with PBS and then treated with ReLeSR (STEMCELL Technologies) for 1–2 min at 37 °C. Alternatively, cells were washed once with PBS and then treated with 2 mM EGTA (SIGMA) for 20 min at 37 °C[47].

**Single-cell passaging**. hESC colonies were dissociated into single cells by adding 1 mL of 0.05% Trypsin-EDTA (Life Technologies) or 1 mL Accutase (Innovative Cell Technologies) to cells in a 9.6 cm² well, incubating cells for 5–7 min at 37 °C, and quenching with 1 mL of ES-qualified FBS (Millipore). Cell clumps were broken up by gently flushing cells 5–10 times with a P1000 micropipette. Afterward, cells were collected, centrifuged at $200 \times g$ for 3 min, and re-suspended in mTeSR1 supplemented with 10 μM ROCK inhibitor. In total, 200,000 to 1,200,000 cells were seeded into a 15 mm well coated with Matrigel.

**Epifluorescence imaging of hESCs**. hESCs were imaged on a Zeiss Axiovision inverted microscope with Zeiss ×10 and ×20 plan apo objectives (NA 1.3) using the appropriate filter sets and an Orca-Flash 4.0 camera (Hamamatsu). The 38 HE GFP/43 HE DsRed/46 HE YFP/47 HE CFP/49 DAPI/50 Cy5 filter sets from Zeiss were used.

**Confocal imaging of hESCs**. Cells were imaged on a Zeiss LSM 700 confocal microscope with Zeiss ×40 and ×63 oil objectives (NA 1.3) with the appropriate filter sets and a back-thinned Hamamatsu EMCCD camera.

**Mouse embryo recovery**. Eight-week-old adult C57BL/6J female mice were naturally mated and sacrificed at 6 a.m. (E6.25), 12 p.m. (E6.5), or 6 p.m. (E6.75) on the sixth day post coitum. In each case, the uterus was recovered, and embryos were dissected from the deciduae[48,49] in embryo culture buffer (see Mouse embryo culture).

**Mouse embryo microinjection**. Embryos were transferred to a microinjection chamber immersed in PBS. These microinjection chambers were made with 0.4% agarose and had multiple channels for holding embryos (Supplementary Fig. 15c). They were specifically designed to minimize the movement and deformation of embryos during microinjection. Microinjection needles were made by pulling glass capillaries (Kwik-Fil, 1B100F-4, World precision instruments) in a micropipette puller (Model P-97, Sutter instrument) using a custom program (Heat 516, Pull 99, Vel 33, and Time 225). The needle was back-filled with 1.5–2.0 μg/μL plasmid purified using an endotoxin-free maxiprep kit (NucleoBond Xtra Maxi Plus EF, 740426.10, Macherey-Nagel). To reduce jamming during microinjection, the plasmid solution was centrifuged at $5000 \times g$ for 10 min, and the supernatant was loaded into the needle. The microinjection needle was inserted into the pre-amniotic cavity, and the plasmid solution was injected using air pressure (Xeno-Works digital microinjector, Sutter instrument) so that the cavity expanded slightly.

**Mouse embryo electroporation**. Microinjected embryos were transferred to the electroporation chamber immersed in PBS (Supplementary Fig. 15c). Electrodes in the chamber were made of 0.127 mm platinum wires (00263, Alfa Aesar). Embryos were placed at the center of the chamber, either parallel or perpendicular to platinum wires. Three electric pulses[50] (30 V, 1 ms duration, 1 s apart) were delivered using a square wave electroporator (ECM 830, BTX).

**Mouse embryo culture**. Electroporated embryos were transferred to a 12-well cell culture dish containing embryo culture media at 37 °C and 5% $CO_2$. This media contains 50% rat serum (AS3061; Valley Biomedical) and 50% Ham's F12 (31765035; Thermo Fisher) supplemented with N-2 (17502048; Thermo Fisher)[51]. The media was equilibrated in the incubator for 1 h prior to embryo addition. E7.5 embryos cultured in this media developed heartbeats after 24–36 h (Video S1). Electroporated embryos were cultured for 4 h before immunostaining. Only embryos without visible defects were subjected to downstream analysis.

**Surface immunostaining of embryos**. The EXE and underlying VE were removed from embryos using fine forceps (1125200; Dumont). The remaining epiblast and VE were incubated in primary antibodies diluted in embryo culture media with 1% BSA and 5% normal donkey serum for 45 min at 37 °C and 5% $CO_2$. The embryos were subsequently washed three times with PBS before being fixed for 30 min at room temperature with 4% formaldehyde. Due to this fixation step, occasionally

aggregates of unbound antibodies were retained inside the pre-amniotic cavity. These large aggregates, having no DAPI or OCT4 stain, were excluded from downstream analysis.

**Intracellular immunostaining of embryos.** Embryos were fixed for 30 min at room temperature in 4% formaldehyde and rinsed three times with PBS. Permeabilization and blocking were performed simultaneously by incubating cells in 5% normal donkey serum, 1% BSA, and 0.3% Triton X-100 in PBS for 1 h at room temperature. Primary antibody incubation was performed overnight with 1% BSA, and 0.3% Triton X-100 in PBS at room temperature. In the morning following primary incubation, embryos were washed three times with PBS and then incubated with secondary antibodies in staining buffer (as above) for 1 h at room temperature. After secondary stain, embryos were washed three times with PBS before imaging.

**Light-sheet imaging of embryos.** Stained embryos were embedded into low-melting agarose (BP165-25; Thermo Fisher) containing 0.1 µm fluorescent beads (F8801; Thermo Fisher). The embedded embryos were then imaged in a Zeiss Light-sheet Z1 microscope under ×20 water objective from four angles. The resulting multi-view images were registered using ImageJ plugin multi-view reconstruction.

**Fabrication of microfluidic devices.** Microfluidic devices were fabricated in poly (dimethylsiloxane) PDMS using rapid prototyping and soft lithography following published procedures[52]. A photomask was designed to create microfluidic devices that generate linear concentration gradients. A 100-µm-thick "negative" master mold was fabricated from the photomask by patterning SU-8 3050 photoresist on an Si wafer through photolithography. "Positive" replicas were generated by molding PDMS against the master. After devices were cured, three inlets and one outlet with 0.5 mm diameters were punched. The mold-side surfaces of devices were rendered hydrophilic by plasma oxidation through a 5 min plasma treatment in room air with a plasma cleaner (Harrick Plasma) at high RF power. Immediately after plasma treatment, devices were submerged in deionized water and autoclaved at 121 °C and 100 kPa for 20 min in liquid cycle to simultaneously sterilize the devices and remove toxic non-cross-linked monomers. Bubbles were removed from the autoclaved devices by vacuum desiccation for 30 min. Afterward, autoclaved Tygon tubing (Saint Gobain) was inserted into inlets and outlets, and the entire device was sterilized again with 30 min of UV light in a Class II Biological Safety Cabinet. For all experiments using the microfluidic devices, the amount of time the microfluidic devices spent not submerged in water or cell culture media after plasma treatment was minimized to maintain the hydrophilicity of the molded surface.

**Culture of hESCs in microfluidic devices.** hESCs to be cultured in microfluidic devices were passaged and maintained in dish culture as described earlier in Methods. At 1 h prior to application of microfluidic devices, cell culture media was changed to mTeSR supplemented with penicillin–streptomycin solution (×100; Sigma Aldrich). Immediately prior to application of microfluidic devices, the tubing of microfluidic devices was filled with mTeSR + penicillin–streptomycin and clamped closed at ends. Devices were then directly attached to the hESC dish using an aluminum clamp custom-designed to fit the dish. Microfluidic devices were positioned with their molded surface over the hESCs and gently clamped downward onto the dish such that cells were located in the cell chamber. Afterward, inlet tubing was connected to media reservoirs containing mTeSR + penicillin–streptomycin, and outlet tubing was connected to a 3 mL syringe loaded on a syringe pump (Harvard Apparatus). The syringe pump was set to withdraw fluid at a flow rate of 20 µL/min or less. The clamped dish was then placed back into an incubator or loaded onto a Zeiss Axiovision inverted microscope for time-lapse imaging, followed by unclamping all attached tubing and starting the syringe pump. After an hour of flow through the microfluidic device to prime the gradient over the cells, the media in reservoirs was changed to the appropriate differentiation conditions either by adding chemicals directly or by progressive dilution. At the end of microfluidic experiments, 1 mg/mL fluorescein isothiocyanate-dextran (Sigma Aldrich) was added to inlet reservoirs to measure the gradient profile within the device. Once a stable gradient was detected and imaged, the microfluidic device was unclamped from the plate, and cells were fixed and immunostained in situ following standard procedures (see Intracellular immunostaining of hESCs).

**Construction of dual-color hESCs.** TALEN genes targeting *POU5F1* (AI-CN330 targeting TCTGGGCTCTCCCAT; AI-CN331 targeting TCCCCCATTCCTAGA AGG) were prepared using the REAL method[53] to match reported target sites[54]. The TALEN genes targeting *SOX2* (AI-CN298 targeting TTAACGGCACACT GCCC; AI-CN299 targeting TCCAGTTCGCTGTCCGGC) were made by the Joung lab (Massachusetts General Hospital) using the FLASH method (PMID: 22484455). *POU5F1* homology-directed repair (HDR) donors AI-CN623 and AI-

CN684 were used for constructing the *POU5F1*^RFP/+ and *SOX2*^YFP/+*POU5F1*^RFP/+ lines, respectively. The *SOX2* HDR donor was AI-CN600.

H1 cells at p38-39 were treated with 1 µM thiazovivin (StemRD) one day prior to electroporation (Neon; Invitrogen; resuspension buffer R; 100 µL electroporation tip; 1050 V, 30 ms pulse width, 2 pulses; 1.5 or 2 × 10^6 cells) as single cells (StemPro Accutase, Life Technologies) with 1.5 or 3 µg of each TALEN plasmid and 6 or 12 µg of the HDR donor plasmid. The cells were treated with 2 µM thiazovivin for 24 h following electroporation. After recovery, cells were treated with 1 µg/mL puromycin (Life Technologies) for 3 days. Following 3 days of recovery, dual *SOX2* and *POU5F1*-targeted cells were treated with 75 µg/mL G418 sulfate (Life Technologies) for 3 days. Fluorescent colonies were validated by PCR (*SOX2* 5′ junction primers: CCTGATTCCAGTTTGCCTCTCTCTTTTTTTC, CTTATC GTCGTCATCCTTGTAATCAGATCTCC; *POU5F1* 5′ junction primers: ATGCTGTTACTCAGCAAGTCCAAAGCTTG, GCGTAGTCTGGGAC GTCGTATGGGTAAG), had normal karyotypes (Cell Line Genetics), and Southern blots (Lofstrand) confirmed insertion of fluorescent protein transgenes at only the targeted loci in *SOX2*^YFP/+*POU5F1*^RFP/+ (AI01e-SOX2OCT4) and *POU5F1*^RFP/+ (AI05e-OCT4RFP). Silencing of SOX2-YFP was occasionally observed in a small fraction of *SOX2*^YFP/+*POU5F1*^RFP/+ cells. This silenced population was regularly removed by fluorescence-activated cell sorting (FACS).

**Time-lapse microscopy.** For live-cell microscopy, a Zeiss Axiovision microscope was enclosed within an environmental chamber in which CO$_2$ and temperature were regulated at 5% and 37 °C, respectively. Time-lapse images were acquired every 10 min for 18–48 h. Image acquisition was controlled by Zen (Zeiss); all cell tracking was manually performed using the TrackMate package in ImageJ (NIH).

**Image analysis of hESCs.** Cell segmentation and fluorescence measurements were done using CellProfiler[55]. All other hESC image data analysis was performed using custom code written in Matlab (MathWorks). Distance from edge ($d_{edge}$) was calculated as the raw distance of a cell from the colony edge normalized by the average cell diameter (13 µm). $P$ values and confidence intervals were determined by paired $t$-test.

**Proficiency calculation.** Segmented cells from microfluidic experiments were binned according to their $d_{source}$, $d_{edge}$, pSMAD1/5 level, OCT4/SOX2 ratio and/or BRA-CHYURY/T level into 6, 3, 2, 4, and 2 bins, respectively. The bins for $d_{edge}$ were $d_{edge}$ < 2, 2 < $d_{edge}$ < 6, and $d_{edge}$ > 6, where $d_{edge}$ is in units of cell widths. Bins for pSMAD1/5 and BRACHYURY/T levels were calculated by fitting the null distribution to a Gaussian distribution and binning cells as less than or greater than 10 standard deviations from the null distribution mean. Bins for $d_{source}$ and OCT4/SOX2 ratios were determined as evenly distributed percentiles of the total data. Our results did not qualitatively vary with the number of bins or binning algorithm. For each binned variable $X$ and each pair of binned variables $X$ and $Y$, the discrete marginal probability distribution $P_x(x)$ and joint probability distribution $P_{(X,Y)}(x,y)$ were calculated from the corresponding bin frequencies. The mutual information between variables $X$ and $Y$ was calculated as $I(X;Y) = \sum_y \sum_x P_{(X,Y)}(x,y) \log\left(\frac{P_{(X,Y)}(x,y)}{P_X(x)P_Y(y)}\right)$, and the entropy of a variable $Y$ was calculated as $H(Y) = -\sum_y P_Y(y) \log P_Y(y)$. The proficiency[32] for $X$ to predict $Y$ (also called the uncertainty coefficient or entropy coefficient) was then calculated as $U(Y|X) = \frac{I(X;Y)}{H(Y)}$. The proficiency can be intuitively understood as the mutual information shared between variables $X$ and $Y$ normalized by the information entropy of $Y$, describing the fraction of bits of information about $Y$ that can be predicted by knowing the value of $X$. Distributions for proficiencies were determined via bootstrapping by resampling cells 10,000 times with replacement.

**Compliance to ethical and other regulations.** We have complied with all relevant ethical regulations for animal testing and research. Our use of animal is approved by Harvard University IACUC (protocol #15-01-229). Our use of human embryonic stem cells is approved by Harvard University IRB (protocol #IRB18-0665) and Harvard University ESCRO (protocol E00065).

**Reporting summary.** Further information on research design is available in the Nature Research Reporting Summary linked to this article.

## Data availability

The authors declare that all data supporting the findings of this study are available within the article and its supplementary information files or from the corresponding author upon reasonable request.

## Code availability

The C-code for simulating BMP4 dynamics in mouse embryo is available at https://doi.org/10.6084/m9.figshare.9684992. The Matlab code for image data analysis is available on FigShare: at https://doi.org/10.6084/m9.figshare.9805298

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

## Acknowledgements

We thank Dr. Gregg Wildenberg for discussions and Dr. Doug Richardson at the Harvard Center for Biological Imaging for technical assistance. We also thank our laboratory and Dr. Xue Fei for comments on the manuscript. This work is supported by 1R01GM131105-01 (SR) and NSF-Simons Center for Mathematical and Statistical Analysis of Biology (DMS-174269).

## Author contributions

Conceptualization, Z.Z. and S.R.; methodology, Z.Z., S.Z. and S.R.; investigation, Z.Z., S.Z., E.L. and S.R.; writing—original draft, Z.Z., S.Z. and S.R.; writing—reviewing and editing, Z.Z., S.Z., and S.R.; funding acquisition, S.R., resources, Z.Z., S.Z., E.L., J.S.G. and S.R.; supervision, S.R.

## Competing interests

The authors declare no competing interests.
