## [Peer Review File · Nature Communications]

Reviewers' Comments:

Reviewer #1:

Remarks to the Author:

This manuscript by Zhang, Zwick, et al. uses a combination of simulations, in vitro, and in vivo experiments to explain the effects of epiblast geometry on morphogen gradient formation. They explain how a gradient can form even in uniform ligand distribution in the lumen, through localisation of receptors to a basal interstitial space and diffusion into this interstitial space at restricted sites only. The results are intuitive from a biophysical point of view, and the strength of the work is in combining basic simulations to make predictions that are then qualitatively verified in human ESCs and mouse epiblast.

My main conclusion reviewing this paper is that it is conceptually interesting and (to me) experimentally impressive. I am more qualified to comment on the mathematical model and will restrict my review to this. Here, major revisions are needed to provide sufficient methodological detail and justification of the model approach (see specific comments below). The authors should also provide more context to help the reader judge the conceptual novelty, as at least in the in vitro case I have heard this idea before that diffusion in basal space under an epithelial layer contributes to gradient formation (appropriate references may exist). Finally, the "entropic buffering" sounds like a nice idea, but without further demonstration it may be best confined to the discussion. Otherwise I think this mostly very nice paper would definitely be worth publishing, in a suitably revised form, as it would contribute to thinking in the field.

specific comments:

1. Please justify the dimensionality of the diffusion calculations, which seem to be in a 2D cross-section of the epiblast embryo. Is the set-up rotationally symmetric, in particular with respect to the gap between ExE and epiblast? If not, does your model only represent one z-section of the embryo? Random walks are recurrent in 2D but not 3D, would you expect this to change your simulation results?
2. The importance of entropic buffering isn't clearly demonstrated. Presumably you would expect increased fluctuations for smaller luminal volumes, or decreased BMP concentrations? Do your results show this? Otherwise it's not clear to me that entropic buffering needs to be evoked, in which case it's best left for the discussion only.
3. Is entropic buffering only relevant if ligand release is from the basolateral side of the ExE cells? Please clarify.
4. What does theory/simulation predict for Fig S5E? is it linear or saturating?
5. Most of the model-experiment comparison seems to be qualitative in nature. Please clarify or comment on to what extent quantitative comparison is possible/desirable.
6. Can you replicate the LTG experiment with the model? (as in Fig S2, but with mixed apical and basolateral receptor cells)
7. Most importantly, there are insufficient details on simulation methods, leaving the reader unable to reproduce the work and uncertain about the model justification. For example, please provide relevant equations, such as how displacement was calculated at every time step (presumably not by the given $x^2=2Dt$ relation). In what way were Langevin dynamics used, rather than Brownian dynamics? The reference by the first author has been cited, but (a) that model has more features than used here, presumably, such as hydrodynamic coupling, and (b) the methods of this paper should be self-contained as far as possible. Please also provide well-documented code for the model simulations.
8. Please comment to what extent this model is effective in nature, i.e. is it representing actual cell, receptor, and ligand molecule numbers as expected in the embryo? Or are the numbers lower but in the same ratio of ligands/receptors? If so, please explain and justify.
9. Please explain further why the release of new ligand is coupled to internalisation of bound receptor? Is this to conveniently keep the total concentration constant in the simulation? Is that justified?
10. Please also provide more method details on the calculation of mutual information/proficiency.

Is this averaged over cells, and do cells close to the edge and source contribute more because of the uneven colony shape? Can you show the proficiency for edge and interior cells separately?

11. The Fig1 model is 1+1 dimensions in interstitial space (one longitudinal and one smaller transverse dimension), but in Fig 5a the schematic is 2+1 dimensions? Were simulations repeated using an in vitro geometry? If not, please justify.

12. Related to the previous point, it is easily calculated analytically what concentration profile to expect from a constant point source (at the edge) in a region of decay (as would apply for the interstitial space with receptors). Can you comment to what extent the particle diffusion simulation was necessary? Is this to capture the low-molecule fluctuations, or the receptor saturation for cells near the edge? What did we learn from this model that a much simpler reaction-diffusion calculation does not tell us? Explaining this would highlight the value of your contribution and increase the intuitive understanding of the reader.

Minor comments:

13. Some grammar issues/typos, eg abstract and discussion, please check throughout

14. How was edge depth measured? Was this measured in distance and then converted to an average cell diameter or did you count cells from the edge?

Reviewer #2:

Remarks to the Author:

Review of "Embryo geometry drives formation of robust signaling gradients through receptor localization" by Z. Zhang et al.

In the interesting paper by Z. Zhang et al., the authors explore the role of BMP receptor localization along the apical-basal axes of cells in the responding epiblast in shaping and the robustness of the morphogen "gradient" that forms. They tackled the questions on the importance of receptor localization (specifically BMPR1A and for the hESCs- BMPR2) using a combination of mathematical modeling, imaging within the mouse epiblast, and an analogous system of hESC colonies in different conditions including, towards the end of the manuscript a focus of hESC colonies with microfluidic-induced gradients of BMP4. Overall, this is an ambitious and intriguing study that has significant flaws in the connections between the model results and the biology of the mouse epiblast and to some extent the hESC colonies that bring the interpretation of the model results into question. Specifically, the disparity that persists between the calculated ranges for the model simulation results and the type of gradients that form versus the images for PSMAD1/5 gradient range in the epiblast call into question the modeling veracity or at the very least the biophysical rates upon which the model is built.

Specific comments follow:

The focus of the paper, from a modeling perspective, is on the establishment of the position and strength of the BMP4 source to pattern mammalian epithelial cells to address the longstanding question that is being investigated in multiple systems including *Drosophila*, mouse, fish, etc- why are receptors for BMP signaling and others in the TGF-beta superfamily sorted along apico-basal axes. One possibility that is that this allows for separation of the pools of ligand-receptor activation and tighter control over gradient formation. To investigate the possibilities a model was developed with differential sorting of receptors along the apical-basal axes of epithelial cells to determine the different expectations for the signaling gradient formation and sensitivity of the gradients to changes in the input levels of morphogen.

Question- Are the models based on relevant physics and reasonable parameter choices? No- while the diffusion coefficient for BMP4 is based off of FCS data for Nodal in an eLife paper for diffusion in zebrafish, the rate is still higher than reported there and on the order of magnitude for diffusion through water which has a viscosity of 1cp-. Diffusion depends strongly on viscosity and is likely

much lower, even in the amniotic cavity, than 60 microns squared per second.

Next, it is assumed that every collision of a BMP4 ligand that comes into contact with the cell surface will bind to a receptor if a receptor is available (that is if the receptor is not bound to another ligand). This is demonstrably false- the measured binding and k_{on} formation rates for BMP4 are quite low. The dissociation constants are low due to very tight binding once they are bound to the receptor, however the measured on rates for binding of a BMP4 receptor to the type 1 or type 2 BMP receptor are nowhere near the diffusion limit of the reaction rate, which is what is implied by an instantaneous binding of the ligand.

Once bound to the receptor the turnover rate is given as 15 minutes. Let's assume that is the average lifetime of the ligand-receptor complex. This means the half-life is on the order of 10 minutes, or the decay constant is on the order of 10^{-3} per second. The citations that give the high rates are for studies in a hepatoma cell line that is not relevant to the epiblast presented in this work. There are much better available estimates for the rates. The rate of $\sim 10^{-3}/\text{sec}$ is an unrealistically high rate for the estimated removal/endocytosis and turnover rate for the BMP ligand bound to receptor. The rate of dpp uptake has been measured in *Drosophila* wing imaginal discs by Kicheva et al. where it was found to be on the order of 10^{-4} per second and the zebrafish embryo where the capture and endocytosis leads to an approximate lifetime of $\sim 10^{-5}$ per second. These high diffusion rates and decay rates with instantaneous receptor binding results in models that produces a very steep gradient and a moving wavefront of complete receptor occupancy- (Figure 1f) and this gradient is not reminiscent of the gradients and data shown in figure 3 f and figure 3 h that show a gradient that patterns a much larger field of cells. The model results therefore are not based on supportive biophysics, and the results do not match the measurements in later figures in the paper, weakening the tests of the hypothesis and the claims of robustness.

Question: is the system robust? The modeling results in figure 1f suggest that the gradient is not robust. The claim is that the gradient shifts by only a few cells, however, when the original gradient only has high bound receptors for 2 cells and the perturbed case covers 6 cells, that is a 300% increase in the number of cells above threshold and I would argue that this is expected and not robust. In fact the expected shift for a 10X increase in source levels would shift the gradient by $x_T = -\ln(T/A)/\lambda$ where T is the threshold, A is the amplitude (the strength of the source) for a point source morphogen given by $M = A \cdot \exp(-\lambda x)$. For a 10 fold increase in the source, the threshold should shift by $\ln(10) \sim$ or 2.3 fold (230%) so because the model is saturating receptor and shifting the gradient to the right- it is actually less robust than a standard decay. This could probably be shown in the model the authors develop if the receptors were not saturating as they do.

Minor point: In the domain- there are 26 cells that are 10 microns wide and they have 2 microns between them- leading to $26 \times 10 + 25 = 314$ microns in total length, yet they are covered by a preamniotic cavity and interstitial space that are 260 microns long. Please clarify.

A biological aspect that was not clear to this reviewer was where is the BMP4 normally secreted? Is it secreted into the pre-amniotic cavity or is it secreted only into the interstitial space? This is quite significant to know what the baseline expectation and model should be. Secretion by the ExE into the interstitial space only. The distinction between secretion is the basis for robustness against normally encountered fluctuations versus robustness against injections into the amniotic cavity of the epiblast.

Reviewer #3:

Remarks to the Author:

The aim of Zhang and co-authors is to understand how a gradient of BMP signalling is established in the developing mouse embryo. To do so they first run computer simulations, the results of which they then attempt to validate in *in vivo* experiments using colonies of human ESCs and cultured mouse embryos. They make a convincing case that it is the compartmentalization of the

extra-cellular space within which BMP ligands are diffusing (i.e. two compartments: the pro-amniotic cavity and the interstitial space between visceral endoderm and epiblast), and the baso-lateral localization of BMP receptors in epiblast cells (i. e. below the tight junctions that prevent direct communication between the pro-amniotic cavity and the interstitial space) that drive the formation of a signalling gradient within the epiblast layer. Using a microfluidic approach to expose hESCs to a steep gradient of BMP+ACTIVIN they further demonstrate that given that the same elements (morphogen exposure, tight junctions, baso-lateral localization of the receptors) are in place, the distance of a cell from the edge of the colony matters more for its cell-fate determination than its distance from the source of the morphogen. Although a recent study using hESC colonies grown on adhesive micropatterns already described the lateral localization of BMP and ACTIVIN receptors and its contribution to the patterning of these colonies (Etoc et al., Dev Cell 2016), the computer simulations of Zhang et al and their investigations in the mouse embryo provide a framework to better understand how morphogen signalling actually takes place in vivo. It is a very interesting study and it provides valuable insights into the critical roles played by the channel between the ExE and the epiblast (it conditions the establishment of a gradient of BMP4 ligands), and the pro-amniotic cavity (it buffers fluctuations in BMP4 production).

There is however a couple of issues that I think the authors should address.

- Could they provide a better explanation of what underlies some of the choices they made for the simulations and how different ones would affect their outcomes?

For example, I am interested by their choice of distributing 2000 BMP receptor molecules evenly between all epiblast cell. What is the basis for this number, and how would higher or lower numbers affect the outcome of the simulations?

More generally, BMP ligands are not the only molecules to bind BMP receptors, nor are BMP receptors the only molecules to bind BMP ligands. It would be useful to provide a more accurate description of the interactions that are actually known to affect BMP signalling at this stage and then to explain why they chose not to take them into consideration when designing their model (irrelevant, negligible or too complicated?).

- I mentioned above the number of BMP receptors per cell as a matter of interest partly because a control seems to be missing in the experiments described in Figure 4. The authors transfect/electroporate mutant (LTG) versions of the BMP receptors, which are no longer restricted to a baso-lateral localization, either in hESCs or in epiblast cells of mouse embryos. These cells, presumably endowed with the capacity to respond to BMP ligands arriving at their apical side, exhibit ectopic pSMAD1/5 activation. But these experiments likely results in these cells having far more receptors than is usual, possibly making them more sensitive to BMP ligands in the interstitial space. To find out whether this is the case it would be useful to transfect/electroporate constructs expressing the native LTA versions of these receptors and assess their impact on SMAD1/5 activation.

Minor point:

- The convention is to present images of egg-cylinder stage embryos upright, anterior to the left and posterior to the right. The authors follow this convention in all their figures except in figure 1, which is a bit disorientating.

Reviewer #1 (Remarks to the Author):

This manuscript by Zhang, Zwick, et al. uses a combination of simulations, *in vitro*, and *in vivo* experiments to explain the effects of epiblast geometry on morphogen gradient formation. They explain how a gradient can form even in uniform ligand distribution in the lumen, through localisation of receptors to a basal interstitial space and diffusion into this interstitial space at restricted sites only. The results are intuitive from a biophysical point of view, and the strength of the work is in combining basic simulations to make predictions that are then qualitatively verified in human ESCs and mouse epiblast.

My main conclusion reviewing this paper is that it is conceptually interesting and (to me) experimentally impressive. I am more qualified to comment on the mathematical model and will restrict my review to this. Here, major revisions are needed to provide sufficient methodological detail and justification of the model approach (see specific comments below). The authors should also provide more context to help the reader judge the conceptual novelty, as at least in the *in vitro* case I have heard this idea before that diffusion in basal space under an epithelial layer contributes to gradient formation (appropriate references may exist).

We thank the reviewer for the advice. We added appropriate references in the introduction (line 30): “Surprisingly, several morphogen receptors have been found to localize to either the apical or basolateral membrane of epithelial tissues^{14,18-22}. Such localization can dramatically affect how the target tissue senses morphogens^{14,18,21}.” We also provided more context to help the reader judge the conceptual novelty in the discussion (line 364): “While receptor localization has been recently shown to regulate morphogen signaling in cells *in vitro*^{14,19,21,26} and in adult tissues¹⁸, this study is the first to our knowledge to identify this effect of receptor localization on morphogen signaling in the developing embryo.”

Finally, the “entropic buffering” sounds like a nice idea, but without further demonstration it may be best confined to the discussion.

In response to this comment we added new supplementary figures to further demonstrate the concept of “entropic buffering,” as described in our response to specific comment #2 below.

Otherwise I think this mostly very nice paper would definitely be worth publishing, in a suitably revised form, as it would contribute to thinking in the field.

We thank the reviewer for this kind comment.
specific comments:

1. Please justify the dimensionality of the diffusion calculations, which seem to be in a 2D cross-section of the epiblast embryo. Is the set-up rotationally symmetric, in particular with respect to the gap between ExE and epiblast? If not, does your model only represent one z-section of the embryo? Random walks are recurrent in 2D but not 3D, would you expect this to change your simulation results?

We thank the reviewer for these comments. We assume that a 2D simulation of a cross-section of the mouse embryo is sufficient to model the relevant BMP signaling dynamics, given that the geometry of the epiblast and the expression pattern of BMP4 are approximately rotationally symmetric around the proximal-distal axis. While there are important differences

between the anterior and posterior hemispheres of the embryo (the anterior visceral endoderm secretes TGF- β inhibitors, for instance), we do not model those effects here.

To ensure this assumption is justified, we performed new 3D simulations in which the ExE and epiblast are rotationally symmetric epithelial tissues. We simulated both cases in which the gap between the ExE and epiblast is rotationally symmetric or limited to the posterior side of the embryo (as our images in Supplementary Fig. 1b possibly suggest). We added Supplementary Fig. 5 to show the results of these simulations. We find that, for any given coordinate along the anteroposterior axis, BMP signaling is restricted to the epiblast edge regardless of whether the gap is rotationally symmetric or not, akin to our 2D simulation results. We refer to this figure in the result section (line 132): “Likewise, the signaling gradient forms regardless of whether the embryo is rotationally symmetric or if the channel between the ExE and the epiblast is present only at the posterior side in the simulation (Supplementary Fig. 5), even though in the latter case the gradient is more prominent at the posterior side.”

2. The importance of entropic buffering isn't clearly demonstrated. Presumably you would expect increased fluctuations for smaller luminal volumes, or decreased BMP concentrations? Do your results show this? Otherwise it's not clear to me that entropic buffering needs to be evoked, in which case it's best left for the discussion only.

We thank the reviewer for these suggestions. In response, we carried out the recommended simulations and further clarified the meaning of entropic buffering in the manuscript.

In the initial manuscript, we showed via simulation that the majority of BMP4 ligands accumulate in the pre-amniotic cavity of the mouse embryo independent of the mechanism of their secretion (Supplementary Fig. 2b). We reference and explain this result in the results section (line 108): “Due to the large volume difference between the pre-amniotic cavity and interstitial space and the channel (between the ExE and epiblast) that connects them, the majority of BMP4 ligands accumulate in the cavity on the apical side of the epiblast (Fig. 1e and Supplementary Fig. 2b). This is an entropic effect: the entropy of BMP4 ligands is maximized when the ligands are uniformly distributed between the pre-amniotic cavity and the interstitial space. In other words, the accumulation of BMP4 ligands in the cavity is driven by the same physical forces that allows ink to diffuse through water and ultimately reach a uniform distribution independent of where ink is dropped initially.”

To demonstrate specifically how this accumulation of ligands in the pre-amniotic cavity buffers the BMP4 signaling gradient in the epiblast, we added a new supplementary figure (Supplementary Fig. 4) which shows how the signaling gradient changes with ligand concentration as the size of pre-amniotic cavity decreases. As the reviewer predicted, we indeed observe larger fluctuations in the depth of the signaling gradient in simulations with smaller luminal volumes. We refer to this figure in the result section (line 122) in support of an entropic buffering effect: “However, as the size of the pre-amniotic cavity is reduced in the simulation, increases in ligand concentration shift the signaling gradient significantly further into the epiblast (Supplementary Fig. 4), demonstrating the buffering effect.” This, together with the computational and experimental evidence that disruption of epithelial integrity or receptor localization heavily perturbs the signaling gradient in the epiblast, argues that the accumulation of ligands in the pre-amniotic cavity is important for the observed stability of the BMP signaling gradient.

In contrast, we also included new supplementary figures (Supplementary Fig. 7, Supplementary Fig. 8, Supplementary Fig. 9) demonstrating that similar changes in other simulation parameters (including the diffusion coefficient, receptor turnover rate, ligand-receptor binding probability, and total ligand and receptor numbers) do not perturb the BMP signaling gradient to the same degree. We refer to these figures in the result section (line 128): “**Variations in other simulation parameters, such as the ligand diffusion coefficient D , probability of interaction between ligand and unbound receptors P_{binding} , and turnover rate of ligand-receptor pairs T_t , do not similarly disrupt the formation of this signaling gradient (Supplementary Fig. 7, Supplementary Fig. 8, Supplementary Fig. 9).**”

3. Is entropic buffering only relevant if ligand release is from the basolateral side of the ExE cells? Please clarify.

We thank the reviewer for this important clarification. To address this comment, we repeated our simulations with varying pre-amniotic cavity size for both cases in which BMP4 ligands are secreted apically or basolaterally from the ExE (Supplementary Fig. 4). We observe a similar variation in signaling gradient depth with changing luminal volume for both cases, arguing that this entropic buffering mechanism is relevant whether ligands are secreted apically or basolaterally. This is in agreement with our simulation prediction that the accumulation of ligands in the pre-amniotic cavity does not vary significantly with the mechanism of ligand secretion (Supplementary Fig. 2).

4. What does theory/simulation predict for Fig S5E? is it linear or saturating?

We thank the reviewer for the question. Our simulation predicts a slightly superlinear relationship between the log of total ligand concentration and the depth of the signaling gradient from the epiblast edge. We added a new supplementary figure (Supplementary Fig. 14) displaying this relationship and comparing it to the results of our experiments in hESCs *in vitro*. We refer to the figure in the result section (line 195): “**This limited change in depth is consistent with our simulation results (Supplementary Fig. 14).**”

5. Most of the model-experiment comparison seems to be qualitative in nature. Please clarify or comment on to what extent quantitative comparison is possible/desirable.

We thank the reviewer for the comment. We agree that quantitative comparison between simulation and experimental results is desirable. However, this is challenging given the uncertainty of key parameters in the simulation. Many of these parameters, such as ligand diffusion coefficients and receptor turnover times, are difficult to measure precisely in the embryo.

To overcome this challenge, we performed simulations over a large range of parameter values, chosen based on measurements in different organisms that are available in the literature. In general, these simulations predict that the formation of a robust BMP signaling gradient at the epiblast edge is not sensitive to the choice of these parameters (Supplementary Fig. 7, Supplementary Fig. 8, Supplementary Fig. 9). In fact, the only parameter variations that our simulations are extremely sensitive to are changes in parameters that disrupt the sequestration of apical ligands and basolateral receptors such as the permeability of tight junctions and localization of receptors (Supplementary Fig. 2c), both of which we validated experimentally. Assuming that the true physiological parameters fall within the wide ranges we considered, we can conclude that basolateral receptor localization and compartmentalized

embryo geometry are essential for the formation of the BMP signaling gradient observed in the mouse embryo. We added an explanation of this strategy in the result section (line 128): “Variations in other simulation parameters, such as the ligand diffusion coefficient D , probability of interaction between ligand and unbound receptors P_{binding} , and turnover rate of ligand-receptor pairs T_t , do not similarly disrupt the formation of this signaling gradient (Supplementary Fig. 7, Supplementary Fig. 8, Supplementary Fig. 9).”

6. Can you replicate the LTG experiment with the model? (as in Fig S2, but with mixed apical and basolateral receptor cells)

We thank the reviewer for the suggestion. In response, we performed simulations in which either wild type or mis-localizing receptors are overexpressed by a few cells in the epiblast. These simulations predict that apical localization of receptors indeed leads to ectopic BMP signaling regardless of distance from epithelial edge, and show that this effect is not a consequence of the changing number of receptors (Supplementary Fig. 10). We added a reference to this figure in the result section (line 140): “Third, the mis-localization of BMP receptors to the apical membrane should lead to ectopic BMP signaling in the epiblast (Supplementary Fig. 10), since apically localized receptors will be able to detect BMP4 ligands that are buffered in the lumen (Fig. 1e).”

7. Most importantly, there are insufficient details on simulation methods, leaving the reader unable to reproduce the work and uncertain about the model justification. For example, please provide relevant equations, such as how displacement was calculated at every time step (presumably not by the given $x^2=2Dt$ relation). In what way were Langevin dynamics used, rather than Brownian dynamics? The reference by the first author has been cited, but (a) that model has more features than used here, presumably, such as hydrodynamic coupling, and (b) the methods of this paper should be self-contained as far as possible. Please also provide well-documented code for the model simulations.

We agree with the reviewer’s comment that further method details would be useful for reader comprehension. We uploaded commented code for the model simulations as SI. We also added significantly more detail to the description of the simulation in the “Simulation of BMP4 dynamics” methods section so that our model is as self-contained, understandable, and reproducible as possible (line 731). For example, we provided relevant equations for how displacement was calculated at every time step (line 749): “After secretion, BMP4 ligand diffusion is simulated as a random walk following Brownian dynamics. Ligand positions are updated after each time step h according to the equation $\vec{r}_i(t+h) = \vec{r}_i(t) + \vec{\Gamma}_i(t)$, where $\vec{r}_i(t)$ is the position of ligand i at time t , $\vec{\Gamma}_i(t)$ is a random Brownian force acting on ligand i that satisfies constraints $\langle \vec{\Gamma}_i(t) \rangle = 0$ and $\langle \vec{\Gamma}_i(t) \vec{\Gamma}_j(t') \rangle = cDh\delta_{ij}\delta_{tt'}$, where D is the diffusion coefficient and $c=4$ for 2D simulations or $c=6$ for 3D simulations²⁵. We estimate $D=20 \mu\text{m}^2/\text{s}$ by default based on diffusion measurements of BMP homolog Dpp in the larval wing disc of *Drosophila melanogaster*¹⁷. We use the “local” diffusion coefficient measured by fluorescence correlation spectroscopy rather than the “global” diffusion coefficient measured by FRAP since our simulation explicitly models ligand-receptor binding, which has been shown to slow ligand diffusion at larger length scales^{2,17}. Each simulation integration step occurs after $h=3 \text{ ms}$.”

8. Please comment to what extent this model is effective in nature, i.e. is it representing

actual cell, receptor, and ligand molecule numbers as expected in the embryo? Or are the numbers lower but in the same ratio of ligands/receptors? If so, please explain and justify.

We thank the reviewer for these important questions. We can estimate the number and size of the cells in the pre-gastrulation mouse embryo from our 3D confocal/light-sheet images, which informed the relevant parameter choices in our simulation. However, we admittedly do not know the actual numbers of ligands and receptors nor their ratio in the mouse embryo. To overcome this limitation, we performed simulations in which the total numbers of ligand and receptors were varied, as well as simulations in which the ratio of ligands to receptors is varied from the $\ll 1$ to the $\gg 1$ regimes (Supplementary Fig. 9). We found that a similar BMP signaling gradient forms at both limits, and that the observed gradient is independent of the total number of ligands and receptors if their ratio is held constant. Assuming the physiological ratio of ligands/receptors is between those limits, our proposed mechanism can be effective in describing BMP signaling in the mouse embryo. We added an explanation of this strategy in the methods section (line 743): “Each epiblast cell has 100 receptors. In 2D simulation, by default 1000 ligands are initially secreted uniformly from the ExE at either the apical membrane or the basal membrane. Although the true number of ligands and receptors may likely be different in the mouse embryo, our simulation results hold for a wide range of scenarios, from the regime where ligands (4000) heavily outnumber receptors (2000) to the regime where receptors (2000) heavily outnumber ligands (200). 3D simulation, in comparison, contains 40000 receptors and 4000-80000 ligands.”

9. Please explain further why the release of new ligand is coupled to internalisation of bound receptor? Is this to conveniently keep the total concentration constant in the simulation? Is that justified?

We thank the reviewer for this clarification. We indeed make the assumption in our simulation that the secretion and receptor-mediated internalization of ligands to conveniently keep the total concentration constant. We initially expected that this assumption should not affect our simulation results in the case where changes in the total ligand concentration are much slower than the time scale of signal transduction (90 min or less). However, it is certainly true that there is no true physiological coupling between these two processes and that the embryo is likely far from equilibrium. To understand if this simplification is permissible to understand the system, we performed additional simulations in which the release of new ligands is not coupled to internalization and recycling of bound receptors. In particular, we examined how the simulation performed at the non-equilibrium regimes in which (1) total ligand concentration steadily decreases after an initial burst of secretion or (2) increases as ligand secretion heavily outpaces receptor-mediated turnover. Holding all other parameters the same, we found that both simulations predicted the formation of signaling gradients at the epiblast edge similar to those in our simplified model after the same period of time (Supplementary Fig. 8). We added a reference to this figure in the result section (line 128): “Variations in other simulation parameters, such as the ligand diffusion coefficient D , probability of interaction between ligand and unbound receptors P_{binding} , and turnover rate of ligand-receptor pairs T_t , do not similarly disrupt the formation of this signaling gradient (Supplementary Fig. 7, Supplementary Fig. 8, Supplementary Fig. 9).”

10. Please also provide more method details on the calculation of mutual information/proficiency. Is this averaged over cells, and do cells close to the edge and source contribute more because of the uneven colony shape? Can you show the proficiency for edge and interior cells separately?

We thank the reviewer for pointing out this omission. In response, we modified Fig. 5b, d, h to include confidence intervals and added “Proficiency calculation” section to the methods (line 1040): “Segmented cells from microfluidic experiments were binned according to their d_{source} , d_{edge} , pSMAD1/5 level, OCT4/SOX2 ratio and/or BRACHYURY/T level into 6, 3, 2, 4, and 2 bins, respectively. The bins for d_{edge} were $d_{\text{edge}} < 2$, $2 < d_{\text{edge}} < 6$, and $d_{\text{edge}} > 6$, where d_{edge} is in units of cell widths. Bins for pSMAD1/5 and BRACHYURY/T levels were calculated by fitting the null distribution to a Gaussian distribution and binning cells as less than or greater than 10 standard deviations from the null distribution mean. Bins for d_{source} and OCT4/SOX2 ratios were determined as evenly distributed percentiles of the total data. Our results did not qualitatively vary with the number of bins or binning algorithm. For each binned variable X and each pair of binned variables X and Y , the discrete marginal probability distribution $P_X(x)$ and joint probability distribution $P_{(X,Y)}(x,y)$ were calculated from the corresponding bin frequencies. The mutual information between variables X and Y was calculated as $I(X;Y) = \sum_y \sum_x P_{(X,Y)}(x,y) \log \left(\frac{P_{(X,Y)}(x,y)}{P_X(x)P_Y(y)} \right)$ and the entropy of a variable Y was calculated as $H(Y) = -\sum_y P_Y(y) \log P_Y(y)$. The proficiency³⁰ for X to predict Y (also called the uncertainty coefficient or entropy coefficient) was then calculated as $U(Y|X) = \frac{I(X;Y)}{H(Y)}$. The proficiency can be intuitively understood as the mutual information shared between variables X and Y normalized by the information entropy of Y , describing the fraction of bits of information about Y that can be predicted by knowing the value of X . Proficiency confidence intervals were determined by the percentiles of bootstrap distributions after resampling cells 10,000 times with replacement.”

As the reviewer astutely noted, some types of cells can contribute more to the proficiency calculation when the probability distributions of cell types are not uniform. For instance, large colonies will have many more interior cells than edge cells, resulting in interior cells being weighted more strongly as a subpopulation. It is evident from the images in Fig. 5e that the distributions of d_{source} and d_{edge} of cells within the microfluidic device are not uniform. We explicitly show these distributions for our microfluidic experiment in Supplementary Fig. 20.

To address these concerns, we resampled cells from our data uniformly across d_{source} , d_{edge} , or both variables simultaneously, resampling the same number of cells with replacement following standard bootstrapping procedures (Supplementary Fig. 20). We repeated this resampling 10,000 times and calculated the respective proficiencies for d_{source} and/or d_{edge} to predict pSMAD1/5, OCT4/SOX2, and BRACHYURY/T values. In all cases, we obtain the same qualitative results, namely (1) d_{source} and d_{edge} carry independent information about cell signaling and fate choice and (2) d_{edge} almost always has a greater proficiency to predict cells’ pSMAD1/5, OCT4/SOX2, and BRACHYURY/T levels than d_{source} . We added a reference to this figure in the result section (line 298): “In fact, a cell’s d_{edge} had a significantly higher proficiency than d_{source} in predicting its signaling response to the BMP gradient (Fig. 5d and Supplementary Fig. 20a-f).”

11. The Fig1 model is 1+1 dimensions in interstitial space (one longitudinal and one smaller transverse dimension), but in Fig 5a the schematic is 2+1 dimensions? Were simulations repeated using an in vitro geometry? If not, please justify.

We thank the reviewer for the suggestion of repeating the simulation using an *in vitro* geometry. We modified our 3D simulations by removing the ExE and organizing the epiblast cells as a flat tissue to mimic the *in vitro* geometry of hESC colonies. All other parameters were kept the same. We added Supplementary Fig. 18 to show the results of these simulation, in which a BMP signaling gradient is predicted to form from edge of hESC colonies treated with a uniform media concentration of BMP4 (as in Fig. 3) or exposed to a BMP4 gradient (as in Fig. 5a). We added a reference to this figure in the result section (line 296): “Consistent with our previous results (Fig. 3, Supplementary Fig. 12a,b, and Supplementary Fig. 18), signaling activity depended strongly on d_{edge} (Fig. 5c,d and Supplementary Fig. 17d).”

12. Related to the previous point, it is easily calculated analytically what concentration profile to expect from a constant point source (at the edge) in a region of decay (as would apply for the interstitial space with receptors). Can you comment to what extent the particle diffusion simulation was necessary? Is this to capture the low-molecule fluctuations, or the receptor saturation for cells near the edge? What did we learn from this model that a much simpler reaction-diffusion calculation does not tell us? Explaining this would highlight the value of your contribution and increase the intuitive understanding of the reader.

We thank the reviewer for this comment. We agree that the reaction-diffusion model is a classical, useful, and intuitive tool in studying morphogen gradient formation and tissue patterning during development. While it is true that such a model of BMP4 signaling in the mouse embryo is almost certainly analytically solvable, we anticipated that the three-dimensional embryonic geometry, receptor localization, and polarized ligand secretion would add boundary conditions and equations to the model that would be more complicated to explain in an interdisciplinary manuscript. Ideally, we would like the main results of our manuscript to be easily understood by a wide variety of readers, including those lacking a quantitative background. Accordingly, we expect that a particle diffusion simulation (including its implications and its limitations) will be intuitively understandable for such readers. In particular, perturbations of the embryonic geometry (Supplementary Fig. 4), receptor localization (Supplementary Fig. 10), and ligand secretion (Supplementary Fig. 2) are easily implemented in the particle diffusion simulation, making the results more easily interpretable. Further, a particle diffusion simulation simplifies the modeling of more complex phenomena that may or may not be easily solvable analytically, including noisy fluctuations or cases when the system is not at equilibrium (Supplementary Fig. 8). We added a justification for using a particle diffusion simulation approach in the method section (line 798): “A particle diffusion simulation was utilized rather than a reaction-diffusion model to study the effects of (i) embryonic geometry (Supplementary Fig. 4), (ii) polarized ligand secretion (Supplementary Fig. 2), (iii) receptor mis-localization (Supplementary Fig. 10) on BMP signaling with an intuitively understandable approach.”

Minor comments:

13. Some grammar issues/typos, eg abstract and discussion, please check throughout

We thank the reviewer for this comment. We have fixed grammar issues and minor inconsistencies throughout the manuscript, particularly in the abstract and discussion.

14. How was edge depth measured? Was this measured in distance and then converted to an average cell diameter or did you count cells from the edge?

We thank the reviewer for this question. We added a description of the edge depth calculation

to the methods (line 1035): “Distance from edge (d_{edge}) was calculated as the raw distance of a cell from the colony edge normalized by the average cell diameter (13 μm).” We represented edge depth in these units (1) for simplicity and clarity for the reader, (2) to compare human and mouse data more naturally as cell sizes are slightly different, (3) because cell diameter did not vary much near colony edges where cells are not tightly confined, and (4) since our results generally varied along distances much greater than 13 μm .

--

Reviewer #2 (Remarks to the Author):

Review of “Embryo geometry drives formation of robust signaling gradients through receptor localization” by Z. Zhang et al.

In the interesting paper by Z. Zhang et al., the authors explore the role of BMP receptor localization along the apical-basal axes of cells in the responding epiblast in shaping and the robustness of the morphogen “gradient” that forms. They tackled the questions on the importance of receptor localization (specifically BMPRI1A and for the hESCs- BMPRI2) using a combination of mathematical modeling, imaging within the mouse epiblast, and an analogous system of hESC colonies in different conditions including, towards the end of the manuscript a focus of hESC colonies with microfluidic-induced gradients of BMP4. Overall, this is an ambitious and intriguing study that has significant flaws in the connections between the model results and the biology of the mouse epiblast and to some extent the hESC colonies that bring the interpretation of the model results into question. Specifically, the disparity that persists between the calculated ranges for the model simulation results and the type of gradients that form versus the images for PSMAD1/5 gradient range in the epiblast call into question the modeling veracity or at the very least the biophysical rates upon which the model is built.

Specific comments follow:

The focus of the paper, from a modeling perspective, is on the establishment of the position and strength of the BMP4 source to pattern mammalian epithelial cells to address the longstanding question that is being investigated in multiple systems including *Drosophila*, mouse, fish, etc- why are receptors for BMP signaling and others in the TGF-beta superfamily sorted along apico-basal axes. One possibility that is that this allows for separation of the pools of ligand-receptor activation and tighter control over gradient formation. To investigate the possibilities a model was developed with differential sorting of receptors along the apical-basal axes of epithelial cells to determine the different expectations for the signaling gradient formation and sensitivity of the gradients to changes in the input levels of morphogen.

Question- Are the models based on relevant physics and reasonable parameter choices? No- while the diffusion coefficient for BMP4 is based off of FCS data for Nodal in an eLife paper for diffusion in zebrafish, the rate is still higher than reported there and on the order of magnitude for diffusion through water which has a viscosity of 1cp-. Diffusion depends strongly on viscosity and is likely much lower, even in the amniotic cavity, than 60 microns squared per second.

We thank the reviewer for this suggested change. In response, we replaced all previous simulation results in the manuscript (e.g. Fig. 1) with new results from simulations in which the BMP4 diffusion coefficient is $D = 20 \mu\text{m}^2/\text{s}$.

Furthermore, we examined how our simulation results change as we vary the diffusion coefficient between 10 to 80 $\mu\text{m}^2/\text{s}$. This range is based on diffusion coefficients for similar morphogens reported in scientific literature: 20 $\mu\text{m}^2/\text{s}$ for DPP, 40-60 $\mu\text{m}^2/\text{s}$ for NODAL, and 50-80 $\mu\text{m}^2/\text{s}$ for FGF8. We specifically consider “local” diffusion coefficients measured by fluorescence correlation spectroscopy (FCS) rather than “global” diffusion coefficient

measured by FRAP for two reasons. First, our simulation models “local” diffusion steps of lengths on the order of sub-microns, a scale that is detectable by FCS but not by FRAP. Second, our simulation explicitly models ligand-receptor binding, which has been shown to slow the diffusion of morphogen at global scales. We added this justification explicitly to the description of the simulation in the methods (line 772): “We estimate $D=20 \mu\text{m}^2/\text{s}$ by default based on diffusion measurements of BMP homolog Dpp in the larval wing disc of *Drosophila melanogaster*¹⁷. We use the “local” diffusion coefficient measured by fluorescence correlation spectroscopy rather than the “global” diffusion coefficient measured by FRAP since our simulation explicitly models ligand-receptor binding, which has been shown to slow ligand diffusion at larger length scales^{2,17}.”

Our simulations show that the BMP signaling gradient forms inward from the epiblast edge over a wide range of diffusion coefficients (Supplementary Fig. 7). Although the precise diffusion coefficient of BMP4 ligands in early post-implantation mammalian embryo is not known, if we assume that the coefficient is comparable to that of analogous morphogens in other model organisms, we can still conclude that this mechanism of gradient formation is relevant in early mammalian embryos. We now refer to these data in the result section (line 128): “Variations in other simulation parameters, such as the ligand diffusion coefficient D , probability of interaction between ligand and unbound receptors P_{binding} , and turnover rate of ligand-receptor pairs T_t , do not similarly disrupt the formation of this signaling gradient (Supplementary Fig. 7, Supplementary Fig. 8, Supplementary Fig. 9).”

Next, it is assumed that every collision of a BMP4 ligand that comes into contact with the cell surface will bind to a receptor if a receptor is available (that is if the receptor is not bound to another ligand). This is demonstrably false- the measured binding and k_{on} formation rates for BMP4 are quite low. The dissociation constants are low due to very tight binding once they are bound to the receptor, however the measured on-rates for binding of a BMP4 receptor to the type 1 or type 2 BMP receptor are nowhere near the diffusion limit of the reaction rate, which is what is implied by an instantaneous binding of the ligand.

We thank reviewer for this comment. The reviewer is correct that ligand-receptor kinetics are quite inefficient. In accordance with their concerns, we introduced a probability parameter, P_{binding} , of interaction between nearby ligands and receptors. We set $P_{\text{binding}}=0.002$ to ensure that the ligand-receptor binding kinetics are not diffusion limited and reproduced all simulation figures in the manuscript. We also added information to the simulation methods section describing this parameter (line 783): “If a ligand contacts an epiblast cell membrane that has any unbound receptors, the ligand binds the receptor with probability $P = P_{\text{binding}}R_{\text{unbound}}$, where P_{binding} is the probability a ligand binds a nearby unbound receptor and R_{unbound} is the fraction of receptors on the membrane that are not bound by ligand. By default, $P_{\text{binding}}=0.002$.”

In addition, we studied how our simulation changes as P_{binding} varies from a regime where the k_{on} rate is not diffusion limited ($P_{\text{binding}}=0.002$) to a regime where the k_{on} rate is certainly diffusion limited like in our previous simulation ($P_{\text{binding}}=1.0$). In all cases, we found that a restricted signaling gradient forms inward from the epiblast edge (Supplementary Fig. 7). As the reviewer predicted, however, the simulated signaling gradient is less steep and covers a larger field of cells when the ligand-receptor binding kinetics are not diffusion limited. We further agree that such a signaling gradient better resembles the true BMP signaling gradient in the early mouse embryo, as we show in new Supplementary Fig.

6. We added a reference to Supplementary Fig. 6 in the result section (line 203): “In harvested mouse embryos stained for pSMAD1/5, we observed a gradient of pSMAD1/5 activity inward from the proximal edges of the epiblast at the pre-streak (~E6.25) through the early streak (~E6.75) stages of development (Fig. 3e,f, Supplementary Fig. 6).” We added a reference to Supplementary Fig. 7 in the result section (line 128): “Variations in other simulation parameters, such as the ligand diffusion coefficient D , the probability of interaction between ligand and unbound receptors P_{binding} , and the turnover rate of ligand-receptor pairs T_t , do not similarly disrupt the formation of this signaling gradient (Supplementary Fig. 7, Supplementary Fig. 8, Supplementary Fig. 9).”

Once bound to the receptor the turnover rate is given as 15 minutes. Let's assume that is the average lifetime of the ligand-receptor complex. This means the half-life is on the order of 10 minutes, or the decay constant is on the order of 10^{-3} per second. The citations that give the high rates are for studies in a hepatoma cell line that is not relevant to the epiblast presented in this work. There are much better available estimates for the rates. The rate of $\sim 10^{-3}$ /sec is an unrealistically high rate for the estimated removal/endocytosis and turnover rate for the BMP ligand bound to receptor. The rate of dpp uptake has been measured in *Drosophila* wing imaginal discs by Kicheva et al. where it was found to be on the order of 10^{-4} per second and the zebrafish embryo where the capture and endocytosis leads to an approximate lifetime of $\sim 10^{-5}$ per second. These high diffusion rates and decay rates with instantaneous receptor binding results in models that produces a very steep gradient and a moving wavefront of complete receptor occupancy- (Figure 1f) and this gradient is not reminiscent of the gradients and data shown in figure 3 f and figure 3 h that show a gradient that patterns a much larger field of cells. The model results therefore are not based on supportive biophysics, and the results do not match the measurements in later figures in the paper, weakening the tests of the hypothesis and the claims of robustness.

We thank reviewer for this suggested change. In accordance, we varied the turnover time in the simulation between 15 minutes to 60 minutes (corresponding to a decay constant on the order of $\sim 10^{-4}$ per second). We indeed found that while a BMP signaling gradient forms from the epiblast edge in simulations over a range of turnover rates, the range of the gradient is larger in simulations with longer turnover times (Supplementary Fig. 7). Therefore, we replaced all simulation results in the main figure with results from simulations with a turnover time of 45 minutes (corresponding to the measured turnover time of Dpp in *Drosophila* wing imaginal discs). We now cite Kicheval, *et al.* in the introduction (line 29): “Many mechanisms have been proposed to explain how morphogens induce signaling gradients in target tissues and therefore direct the spatial organization of cell fates^{1,2,8-18}”, and in the method section (line 770): “After $T_t=45$ minutes, a timescale related to the endocytosis and recycling of ligand-bound receptors^{17,42,43}, each receptor-ligand pair is replaced by an unbound receptor on the same epiblast cell membrane and an unbound ligand secreted by the same ExE cell.” We refer to Supplementary Fig. 7 in the result section (line 128) as mentioned previously: “Variations in other simulation parameters, such as the ligand diffusion coefficient D , the probability of interaction between ligand and unbound receptors P_{binding} , and the turnover rate of ligand-receptor pairs T_t , do not similarly disrupt the formation of this signaling gradient (Supplementary Fig. 7, Supplementary Fig. 8, Supplementary Fig. 9).”

Question: is the system robust? The modeling results in figure 1f suggest that the gradient is not robust. The claim is that the gradient shifts by only a few cells, however, when the original gradient only has high bound receptors for 2 cells and the perturbed case covers 6

cells, that is a 300% increase in the number of cells above threshold and I would argue that this is expected and not robust. In fact the expected shift for a 10X increase in source levels would shift the gradient by $x_T = -\ln(T/A)/\lambda$ where T is the threshold, A is the amplitude (the strength of the source) for a point source morphogen given by $M = A \cdot \exp(-\lambda x)$. For a 10 fold increase in the source, the threshold should shift by $\ln(10) \sim$ or 2.3 fold (230%) so because the model is saturating receptor and shifting the gradient to the right it is actually less robust than a standard decay. This could probably be shown in the model the authors develop if the receptors were not saturating as they do.

We thank the reviewer for identifying this important distinction that we hope to clarify. We show computationally and experimentally that as a consequence of (i) the compartmentalization of the embryo, (ii) the impermeability of tight junctions, and (iii) the localization of receptors, BMP signaling in the mouse embryo is automatically restricted to the edge of the epiblast. This occurs without any inclusion of inhibitors or signaling feedback to the model. This gradient is “robust” in the sense that it remains restricted to the epiblast edge after major perturbations computationally and experimentally, including simulations where we vary important parameters (Supplementary Fig. 7, Supplementary Fig. 8, Supplementary Fig. 9) or experiments where we provide BMP4 ligands directly into the pre-amniotic cavity (Fig. 3g, h). However, we show that perturbations that disturb any of those 3 preconditions, such as the mis-localization of BMP receptors, is sufficient *in silico* and *in vivo* to drastically disrupt the signaling gradient and elicit ectopic BMP signaling far from the source of BMP4.

We specified that the “robustness” is with respect to formation of the gradient in the introduction (line 72): “This entropic buffering renders the formation of BMP signaling gradient robust to fluctuations in BMP4 level”, and result section (line 106): “The basolateral localization of BMP receptors, in conjunction with the asymmetric compartmentalization of the embryo, also makes formation of this BMP signaling gradient robust to fluctuations in the BMP4 source strength”, (line 139): “Second, formation of this signaling gradient will be robust to fluctuations in BMP concentration (Fig. 1f)”, (line 191): “The formation of these BMP signaling gradients were robust to changes in ligand concentration”.

While we provide new computational evidence that the BMP signaling gradient is buffered by the entropic accumulation of ligands in the pre-amniotic cavity (Supplementary Fig. 4), the reviewer is correct in pointing out that the signaling gradient may not be as robust to fluctuations as in standard reaction diffusion models that incorporate negative feedback mechanisms, such as the morphogen-induced expression of its own inhibitors. Such mechanisms have been previously incorporated into models of BMP signaling¹⁴ to confer further robustness to the predicted signaling gradient against fluctuations in morphogen concentration and can be similarly incorporated in our model. Nevertheless, our results show that any model of BMP signaling in the mouse embryo must consider the geometric constraints outlined above in order to describe the experimental data accurately, including how any cofactors and inhibitors are themselves compartmentalized by the embryo.

We attempted to clarify this point in the discussion (line 366): “Our model neglects the possible effects of other regulators of BMP signaling, such as BMP activators and inhibitors. In particular, TGF- β family inhibitors LEFTY1 and CER1 are expressed in the anterior visceral endoderm of the mouse embryo at E5.75 where they are required for proper patterning during gastrulation³³. We anticipate that inclusion of such regulators to the model would restrict BMP signaling more to the posterior edge of the epiblast and could contribute

further robustness to the BMP signaling gradient against fluctuations in ligand concentration^{5,6,9,11,13,14,34}. Nevertheless, our results show that embryonic geometry and receptor localization are sufficient to produce robust gradients of BMP signaling and to explain how mis-localization of BMP receptors leads to ectopic signaling in anterior and distal epiblast cells (Fig. 4). It would be particularly interesting to incorporate BMP regulators in future versions of the model given that they too can be constrained by the compartmentalization of the embryo^{14,35}.”

Minor point: In the domain- there are 26 cells that are 10 microns wide and they have 2 microns between them- leading to $260+54=314$ microns in total length, yet they are covered by a preamniotic cavity and interstitial space that are 260 microns long. Please clarify.

We thank the reviewer for the chance to clarify this issue. The cell width of 10 microns include 2 microns of lateral separation. We revised the methods section (line 754) to explain this: “Each cell is 8 μm wide and 18 μm tall. The pre-amniotic cavity above the cells is 260 μm wide and 30 μm tall. The interstitial space is 260 μm wide and 2 μm tall. The lateral separation between cells is 2 μm .”

A biological aspect that was not clear to this reviewer was where is the BMP4 normally secreted? Is it secreted into the pre-amniotic cavity or is it secreted only into the interstitial space? This is quite significant to know what the baseline expectation and model should be. Secretion by the ExE into the interstitial space only. The distinction between secretion is the basis for robustness against normally encountered fluctuations versus robustness against injections into the amniotic cavity of the epiblast.

We thank the reviewer for the question and suggestion. To our knowledge, there is no conclusive evidence of whether BMP4 secretion is polarized in the mouse ExE. In accordance with the reviewer’s question, we transiently expressed GFP-BMP4 in hESCs to examine their localization prior to secretion. We found that BMP4 accumulated predominately at the apical membrane (Supplementary Fig. 3), consistent with the evidence for the apical secretion of TGF- β ligands in MDCK cells (a classic polarized cell line). We included a reference to these data in the result section (Line 81): “Given the evidence of polarized ligand secretion by epithelial cells *in vitro*^{18,19} (Supplementary Fig. 3)”.

Although this preliminary evidence suggests that BMP4 ligands may be secreted from the mouse ExE apically into the pre-amniotic cavity, we also find that our simulation results are independent of whether the ligands are secreted apically or basolaterally. In particular, we find that approximately the same fraction of ligands should accumulate in pre-amniotic cavity due to entropy, no matter which compartment they are initially secreted into (Supplementary Fig. 2b). As a result, similar signaling gradients form inward from the epiblast edge in both cases (Supplementary Fig. 2a,c). We furthermore reproduced our results in which the pre-amniotic cavity volume is varied for each case of polarized ligand secretion, obtaining similar results for both (Supplementary Fig. 4). We emphasized these results in the result section (line 102): “The signaling gradient forms regardless of whether BMP4 ligands are secreted from the apical or basolateral membrane of the ExE and arises even if ligands are imposed to be uniformly distributed in the pre-amniotic cavity (Supplementary Fig. 2a)””; and (line 115): “Consistently, BMP4 ligands accumulate in the pre-amniotic cavity, regardless of whether the ligands are secreted apically or basolaterally from the ExE in the simulation (Fig. 1e and Supplementary Fig. 2b).”

--

Reviewer #3 (Remarks to the Author):

The aim of Zhang and co-authors is to understand how a gradient of BMP signalling is established in the developing mouse embryo. To do so they first run computer simulations, the results of which they then attempt to validate in in vivo experiments using colonies of human ESCs and cultured mouse embryos. They make a convincing case that it is the compartmentalization of the extra-cellular space within which BMP ligands are diffusing (i.e. two compartments: the pro-amniotic cavity and the interstitial space between visceral endoderm and epiblast), and the baso-lateral localization of BMP receptors in epiblast cells (i.e. below the tight junctions that prevent direct communication between the pro-amniotic cavity and the interstitial space) that drive the formation of a signalling gradient within the epiblast layer. Using a microfluidic approach to expose hESCs to a steep gradient of BMP+ACTIVIN they further demonstrate that given that the same elements (morphogen exposure, tight junctions, baso-lateral localization of the receptors) are in place, the distance of a cell from the edge of the colony matters more for its cell-fate determination than its distance from the source of the morphogen. Although a recent study using hESC colonies grown on adhesive micropatterns already described the lateral localization of BMP and ACTIVIN receptors and its contribution to the patterning of these colonies (Etoc et al., Dev Cell 2016), the computer simulations of Zhang et al and their investigations in the mouse embryo provide a framework to better understand how morphogen signalling actually takes place in vivo. It is a very interesting study and it provides valuable insights into the critical roles played by the channel between the ExE and the epiblast (it conditions the establishment of a gradient of BMP4 ligands), and the pro-amniotic cavity (it buffers fluctuations in BMP4 production).

There is however a couple of issues that I think the authors should address.

- Could they provide a better explanation of what underlies some of the choices they made for the simulations and how different ones would affect their outcomes?

For example, I am interested by their choice of distributing 2000 BMP receptor molecules evenly between all epiblast cell. What is the basis for this number, and how would higher or lower numbers affect the outcome of the simulations?

We thank the reviewer for these questions. Neither the actual numbers of receptors per cell in the epiblast nor the number of BMP4 ligands in the mouse embryo are known. To overcome this uncertainty, we first performed simulations in three different regimes: (i) there are far more receptors than ligands, (ii) the number of receptors is comparable to the number of ligands, and (iii) there are far more ligands than receptors. Our simulation predicts that similar signaling gradients form inward from the epiblast edge in all three scenarios (Supplementary Fig. 9). Furthermore, we find that, if the ligand-receptor ratio is held fixed, changing the absolute number of ligands and receptors results in almost identical signaling gradients in our simulation. Thus, assuming that the physiological ratio of ligands/receptors lies in between our tested values, our simulations suggest that our proposed mechanism can effectively describe BMP signaling in the mouse embryo even when the true parameters are not known.

We added explanation of this strategy in the method section (line 761): “Each epiblast cell has 100 receptors. In 2D simulation, by default 200-4000 ligands are initially secreted uniformly from the ExE at either the apical membrane or the basal membrane. Although the true number of ligands and receptors may likely be different in the mouse embryo, our simulation results hold for a wide range of scenarios, from the regime where ligands (4000)

heavily outnumber receptors (2000) to the regime where receptors (2000) heavily outnumber ligands (200). 3D simulation, in comparison, contains 40000 receptors and 4000-80000 ligands.”

We applied a similar strategy to other parameters in the simulation. By varying the values of parameters including (i) the diffusion coefficient of ligands, (ii) binding probability between ligands and unbound receptors, (iii) endocytosis rate of ligand-bound receptors, (iv) asymmetry of gap between the ExE and epiblast, and (v) polarized secretion of ligands, we show that the proposed mechanism of gradient formation is applicable to wide range of parameters (Supplementary Fig. 7, Supplementary Fig. 8, Supplementary Fig. 9). We describe this explicitly in the result section (line 128): “Variations in other simulation parameters, such as the ligand diffusion coefficient D , the probability of interaction between ligand and unbound receptors P_{binding} , and the turnover rate of ligand-receptor pairs T_t , do not similarly disrupt the formation of this signaling gradient (Supplementary Fig. 7, Supplementary Fig. 8, Supplementary Fig. 9). Likewise, the signaling gradient forms regardless of whether the embryo is rotationally symmetric or if the channel between the ExE and the epiblast is present only at the posterior side in the simulation (Supplementary Fig. 5).”

More generally, BMP ligands are not the only molecules to bind BMP receptors, nor are BMP receptors the only molecules to bind BMP ligands. It would be useful to provide a more accurate description of the interactions that are actually known to affect BMP signalling at this stage and then to explain why they chose not to take them into consideration when designing their model (irrelevant, negligible or too complicated?)

We thank the reviewer for the suggestion. As the referee correctly points out, there are many known extracellular regulators of BMP signaling our current version of the simulation does not incorporate. These may include other TGF- β family ligands that promiscuously bind to BMP receptors, BMP inhibitors such as NOGGIN and CHORDIN, BMP activators such as FURIN or PACE4, and other membrane associated molecules such as collagen and glypican.

While the regulation of BMP signaling by such extracellular regulators in the context of a compartmentalized embryo geometry is a topic of great interest for future studies, we do not include these regulators in the current study because these factors were not necessary to explain the observed experimental data *in vitro* and *in vivo*. In particular, we show that receptor localization and embryonic geometry are sufficient to explain how in our technically challenging experiments, the mis-localization of BMP receptors elicits ectopic signaling in cells in the distal and anterior epiblast, where TGF- β were previously thought to restrict signaling. Given this supporting evidence, we further demonstrate the possible consequences of receptor localization and embryonic geometry for the BMP signaling gradient in the epiblast, including its entropic buffering by the accumulation of ligands in the pre-amniotic cavity. Given previous studies of the role of BMP inhibitors, we expect that their inclusion in the model would only further enhance the robustness of the gradient to fluctuations in morphogen concentration.

To explain this to the reader, we added following justification to the discussion section (line 367): “Our current model neglects the possible effects of other regulators of BMP signaling, such as BMP activators and inhibitors. In particular, TGF- β family inhibitors LEFTY1 and CER1 are expressed in the anterior visceral endoderm of the mouse embryo at E5.75 where they are required for proper patterning during gastrulation³³. We anticipate that inclusion of

such regulators to the model would restrict BMP signaling more to the posterior edge of the epiblast and could contribute further robustness to the BMP signaling gradient against fluctuations in ligand concentration^{5,6,9,11,13,14,34}. Nevertheless, our results show that embryonic geometry and receptor localization are sufficient to produce robust gradients of BMP signaling and to explain how mis-localization of BMP receptors leads to ectopic signaling in anterior and distal epiblast cells (Fig. 4). It would be particularly interesting to incorporate BMP regulators in future versions of the model given that they too can be constrained by the compartmentalization of the embryo³⁵.”

- I mentioned above the number of BMP receptors per cell as a matter of interest partly because a control seems to be missing in the experiments described in Figure 4. The authors transfect/electroporate mutant (LTG) versions of the BMP receptors, which are no longer restricted to a baso-lateral localization, either in hESCs or in epiblast cells of mouse embryos. These cells, presumably endowed with the capacity to respond to BMP ligands arriving at their apical side, exhibit ectopic pSMAD1/5 activation. But these experiments likely results in these cells having far more receptors than is usual, possibly making them more sensitive to BMP ligands in the interstitial space. To find out whether this is the case it would be useful to transfect/electroporate constructs expressing the native LTA versions of these receptors and assess their impact on SMAD1/5 activation.

We thank the reviewer for this suggestion. In accordance, we performed an additional control experiment in which we transfected hESCs with a plasmid containing both wild type BMP receptors. We found that the overexpression of these basolaterally localizing receptors did not lead to ectopic pSMAD1/5 activation (Supplementary Fig. 16). This result is consistent with an additional simulation prediction that a tenfold overexpression of basolaterally localizing BMP receptors should not lead to increased BMP response (Supplementary Fig. 10). We added a reference to this control into the result section (line 243): “**In contrast, overexpression of wild-type receptors did not lead to a comparable increase in pSMAD1/5 levels of transfected cells *in vitro*, as predicted by our simulation (Supplementary Fig. 10, Supplementary Fig. 16).**”

Minor point:

- The convention is to present images of egg-cylinder stage embryos upright, anterior to the left and posterior to the right. The authors follow this convention in all their figures except in figure 1, which is a bit disorientating.

We thank reviewer for this suggestion. We changed the orientation of the mouse embryo in Fig. 1 to conform with the conventional depiction.

Reviewers' Comments:

Reviewer #1:

Remarks to the Author:

The reviewers have made substantial revisions to the manuscript, especially in the form of additional model simulations for parameter sensitivity analysis. They have addressed most of my comments very well. Three issues remain for me:

1. The comparison between experiment and simulation remains somewhat vague. It's ok to have a qualitative comparison if quantitative agreement is not possible based on current measurement techniques, but it's not clear what standards the authors themselves hold their model to. For example, comparing Fig. 1 d and f with Fig. 3 b and d, it seems that the trends in the model for increasing time and increasing concentration are exactly opposite to what is observed in vivo. This isn't helped by the fact that the grey shading in the legend is darker for lower values in Fig 1, and darker for higher values in Fig 3. Indeed this makes me wonder if there is a typo in Fig 1, given the ordering was opposite in the first version of their manuscript.

2. The authors have clarified well in their response addressing the concerns of reviewer 2 regarding the robustness of gradient formation, but in the revised text they could be more explicit still that they think the formation of the gradient is robust, while the size (or length) of the gradient isn't (if I understood correctly).

3. It is good that the authors have provided their simulation code, but they have provided no instructions how to run it. Trying the standard approach for C-code, I got the following compilation errors (on Mac OSX):

```
test.c:56:13: error: initializer element is not a compile-time constant
double *PT3=(double *)malloc(3*WRC*LN*sizeof(double));
^ ~~~~~
test.c:58:10: error: initializer element is not a compile-time constant
int *ST3=(int *)malloc(WRC*LN*sizeof(int));
^ ~~~~~
test.c:60:10: error: initializer element is not a compile-time constant
int *RC3=(int *)malloc(4*WRC*NRC*sizeof(int));
^ ~~~~~
test.c:62:11: error: initializer element is not a compile-time constant
long *RT3=(long *)malloc(3*WRC*LN*sizeof(long));
```

Reviewer #2:

Remarks to the Author:

Overall, the paper and the modeling correspondence with the data presented later in the paper is much improved and the authors have addressed many of the critiques quite well in the revised manuscript. A few items remain:

1. Robustness is used to describe the effect of entropic buffering by the contribution of a large pool of ligand as the source in the pre-amniotic cavity- robustness is not precisely defined and a revision that states more clearly what is meant by robustness would address this. Currently-robustness seems defined in different ways- first, abolishing receptor localization abolishes the gradient- because now all cells are accessing the large pool of ligand in the pre-amniotic cavity- in this definition, robustness is the ability to form a gradient at all vs. robustness as being defined sensitivity to fluctuations vs. parametric sensitivity. Another definition is related to small gradient changes when input concentration is changed over a wide range (1000-fold for hESC).

2. The use of "entropic buffering" is somewhat ambiguous still and, in this reviewer's opinion, unnecessary jargon- the main point is that having a relatively large volume for ligand to

accumulate provides a ligand reservoir that reduces the fluctuations of the input concentration to the patterning of the epiblast. Perhaps “stable reservoir” of ligand is less “jargon” and reflective of the main conclusion. A comparison of volume of the preamniotic cavity vs. the volume VE where the ligand ultimately forms a gradient would be a useful calculation- the preamniotic cavity simply serves as a large ligand reservoir and the pore between EXE and embryo is the channel that allows flow of ligand into the patterning system from the large and stable reservoir.

Reviewer #3:

Remarks to the Author:

I am fully satisfied with the reply of the authors to the comments I made, and also to the issues raised by the other reviewers and that I could understand. I support the publication of this revised manuscript.

Reviewers' comments:

Reviewer #1 (Remarks to the Author):

The reviewers have made substantial revisions to the manuscript, especially in the form of additional model simulations for parameter sensitivity analysis. They have addressed most of my comments very well. Three issues remain for me:

1. The comparison between experiment and simulation remains somewhat vague. It's ok to have a qualitative comparison if quantitative agreement is not possible based on current measurement techniques, but it's not clear what standards the authors themselves hold their model to. For example, comparing Fig. 1 d and f with Fig. 3 b and d, it seems that the trends in the model for increasing time and increasing concentration are exactly opposite to what is observed *in vivo*. This isn't helped by the fact that the grey shading in the legend is darker for lower values in Fig 1, and darker for higher values in Fig 3. Indeed this makes me wonder if there is a typo in Fig 1, given the ordering was opposite in the first version of their manuscript.

We thank the reviewer for pointing out the error in Fig. 1 legend. We corrected the legend, so that darker curve in Fig. 1d corresponds to later time and darker curve in Fig. 1f corresponds to higher concentration. Therefore, the trends in the model for increasing time and increasing concentration are consistent with what is observed *in vitro* and *in vivo* (Fig. 3).

In addition, we expect our model to be consistent with experiments: (i) when tight junctions are broken (Supp Fig. 2c and 13c,d), (ii) when receptors are mis-localized (Fig. 4, Supp Fig. 2c, and Supp Fig. 10), (iii) in mutation information between pSMAD1/5 and distance from epithelial edge (Fig. 5b,d).

We added these standards to the method section (line 813): “Although direct quantitative comparison between the model and experiment is not possible without precise knowledge of biochemical parameters, we expect our model to agree with experiment qualitatively in the following five criteria: (i) pSmad1/5 as a function of time (Fig. 1d and Fig. 3b,f), (ii) pSmad/15 as function of concentration (Fig. 1f, Fig. 3d,h, and Supp Fig. 14), (iii) when tight junctions are broken (Supp Fig. 2c and 13c,d), (iv) when receptors are mis-localized (Fig. 4, Supp Fig. 2c and Supp Fig. 10), (v) mutation information between pSMAD1/5 and distance from epithelial edge (Fig. 5b,d).”

2. The authors have clarified well in their response addressing the concerns of reviewer 2 regarding the robustness of gradient formation, but in the revised text they could be more explicit still that they think the formation of the gradient is robust, while the size (or length) of the gradient isn't (if I understood correctly).

We thank the reviewer for the suggestion. We added the following statement to the result section (line 135): “our simulations demonstrate that the formation of the signaling gradient is robust in that it can form under wide variety of conditions; and further, while the scale of the gradient increases with source strength, this increase is limited by basolateral receptor localization and the asymmetric compartmentalization of the embryo”.

3. It is good that the authors have provided their simulation code, but they have provided no

instructions how to run it. Trying the standard approach for C-code, I got the following compilation errors (on Mac OSX):

```
test.c:56:13: error: initializer element is not a compile-time constant
double *PT3=(double *)malloc(3*WRC*LN*sizeof(double));
^
```

```
test.c:58:10: error: initializer element is not a compile-time constant
int *ST3=(int *)malloc(WRC*LN*sizeof(int));
^
```

```
test.c:60:10: error: initializer element is not a compile-time constant
int *RC3=(int *)malloc(4*WRC*NRC*sizeof(int));
^
```

```
test.c:62:11: error: initializer element is not a compile-time constant
long *RT3=(long *)malloc(3*WRC*LN*sizeof(long));
^
```

We thank the reviewer for requesting code-running instruction. This is what we usually do: (1) compile the C-code in terminal (on Mac OSX or cluster) by: `g++ -o test.exe comment2-loop-simulp3d7tov.cpp`. (2) run the exe by: `./test.exe`. It would perform 10 independent simulations of 5000 ligands. It would take more than 24 hours to complete. One can, however, perform a quicker test run by reducing number of ligands (line 11) or number of trajectories (line 31). We added this instruction to beginning of the C-code, and also provided output files of the test run as SI.

--

Reviewer #2 (Remarks to the Author):

Overall, the paper and the modeling correspondence with the data presented later in the paper is much improved and the authors have addressed many of the critiques quite well in the revised manuscript. A few items remain:

1. Robustness is used to describe the effect of entropic buffering by the contribution of a large pool of ligand as the source in the pre-amniotic cavity- robustness is not precisely defined and a revision that states more clearly what is meant by robustness would address this. Currently- robustness seems defined in different ways- first, abolishing receptor localization abolishes the gradient- because now all cells are accessing the large pool of ligand in the pre-amniotic cavity- in this definition, robustness is the ability to form a gradient at all vs. robustness as being defined sensitivity to fluctuations vs. parametric sensitivity. Another definition is related to small gradient changes when input concentration is changed over a wide range (1000-fold for hESC).

We thank the reviewer for the suggestion. We added the following statement to the result section (line 135): “our simulations demonstrate that the formation of the signaling gradient is robust in that it can form under wide variety of conditions; and further, while the scale of the gradient increases with source strength, this increase is limited by basolateral receptor localization and the asymmetric compartmentalization of the embryo”.

2. The use of “entropic buffering” is somewhat ambiguous still and, in this reviewer’s opinion, unnecessary jargon- the main point is that having a relatively large volume for ligand to accumulate provides a ligand reservoir that reduces the fluctuations of the input concentration to the patterning of the epiblast. Perhaps “stable reservoir” of ligand is less “jargon” and

reflective of the main conclusion. A comparison of volume of the preamniotic cavity vs. the volume VE where the ligand ultimately forms a gradient would be a useful calculation- the preamniotic cavity simply serves as a large ligand reservoir and the pore between EXE and embryo is the channel that allows flow of ligand into the patterning system from the large and stable reservoir.

We agree with reviewer that “stable reservoir” is an accurate description of pre-amniotic cavity and our aim is not to introduce jargon. However, the term “entropic buffering”, explains not only how such “stable reservoir” is formed (entropy) irrespective of whether the ligands are secreted apically or basolaterally, but also why such “stable reservoir” is essential (buffering). Entropy is an essential part of the buffering because there are many more paths for ligands to accumulate and stay in the larger apical cavity than to enter and diffuse through the smaller interstitial space. Besides, given the omnipresence of epithelial tissues in embryos, such “entropic buffering” can be relevant in many other developmental contexts. Therefore, we cannot think of a more precise term for this phenomenon and expect “entropic buffering” to be an important concept for the developmental biology in future. We therefore leave this term in the text as it is.

To better compare the volume of pre-amniotic cavity and interstitial space, we revised Fig. 1c and Supp Fig. 4 so that not only the absolute size of but also the ratio between interstitial space and pre-amniotic cavity are shown.

--

Reviewer #3 (Remarks to the Author):

I am fully satisfied with the reply of the authors to the comments I made, and also to the issues raised by the other reviewers and that I could understand. I support the publication of this revised manuscript.

Reviewers' Comments:

Reviewer #1:

Remarks to the Author:

The authors have addressed my remaining concerns and using the updated instructions I was also able to run the simulation code.